



# A machine learning based global sea-surface iodide distribution

Tomás Sherwen[1,2], Rosie J. Chance[2], Liselotte Tinel[2], Daniel Ellis[2], Mat J. Evans[1,2], and Lucy J. Carpenter[2]

[1]National Centre for Atmospheric Science, University of York, York, YO10 5DD, UK
[2]Wolfson Atmospheric Chemistry Laboratories, University of York, York, YO10 5DD, UK

**Correspondence:** Tomás Sherwen (tomas.sherwen@york.ac.uk)

**Abstract.** Iodide in the sea-surface plays an important role in the Earth system. It modulates the oxidising capacity of the troposphere and provides iodine to terrestrial ecosystems. However, our understanding of its distribution is limited due to a paucity of observations. Previous efforts to generate global distributions have generally fitted sea-surface iodide observations to relatively simple functions of sea-surface temperature (Chance et al., 2014; MacDonald et al., 2014). This approach fails to

account for coastal influences and variation in the bio-geochemical environment. Here we use a machine learning regression approach (Random Forest Regression) to generate a high resolution (0.125°×0.125°, ~12.5 km), monthly dataset of present-day global sea-surface iodide. We use a compilation of iodide observations (Chance et al., 2019b) that is 45% larger than has been used previously (Chance et al., 2014) as the dependent variable and co-located ancillary parameters (temperature, nitrate, phosphate, salinity, shortwave radiation, topographic depth, mixed layer depth, and chlorophyll-a) from global climatologies as

the independent variables. We investigate the regression models generated using different combinations of ancillary parameters and select the ten best-performing models to be included in an ensemble prediction. We then use this ensemble of models, combined with global fields of the ancillary parameters, to predict a new high resolution global sea-surface iodide field. Sea-surface temperature is the most important variable in all of the top ten models. We estimate a global average sea-surface iodide concentration of 106 nM (with an uncertainty of ~20 %), which is within the range of previous estimates (60-130 nM).

Similar to previous work, higher concentrations are predicted for the tropics than for the extra-tropics. Unlike the previous parameterisations, higher concentrations are also predicted for shallow areas such as coastal regions and the South China Sea. Compared to previous work, the new parameterisation better captures observed variability. The iodide concentrations calculated here are significantly higher (40% on a global basis) than the commonly used MacDonald et al. (2014) parameterisation, with implications for our understanding of iodine in the atmosphere. The global iodide dataset is made freely available to the

community (Sherwen et al. (2019); DOI:https://doi.org/10/gfv5v3) and as new observations are made, we will update the global dataset through a "living data" model.



# 1 Introduction

Iodine in seawater exists in two major forms, iodide (I$^-$) and iodate (IO$_3^-$). Total inorganic iodine (I$^-$ + IO$_3^-$) remains approximately constant across most of the oceans, but the ratio of iodide to iodate varies has been shown to vary by Chance et al. (2014) with latitude, depth and oxygen level . A small amount of iodine (<10%) is thought to be present in organic forms in

the open ocean (e.g. Wong (1991)), however, this may be a larger fraction in coastal waters (e.g. Wong and Cheng (1998)). The processes controlling the distribution of the ratio between iodide and iodate remain poorly understood (Chance et al., 2014).

A reason for gaps in our understanding is that the observational dataset of iodide and iodate remains relatively sparse (Chance et al., 2014, 2019b). Despite this paucity in observations, iodine's role in the Earth system has driven multidisciplinary interest in the distribution of iodine compounds in seawater from a number of different research communities, including paleoceanog-

raphy (Lu et al., 2016, 2018; Zhou et al., 2015), atmospheric composition (Saiz-Lopez et al., 2014; Sherwen et al., 2016a), and air-quality prediction (Sarwar et al., 2015; Luhar et al., 2017, 2018).

The atmospheric science community has seen a particularly large growth in interest in iodine chemistry in the atmosphere and at the sea-surface, as sea-surface I$^-$ is believed to be the main driver of atmospheric iodine emissions. The reaction of I$^-$ with ozone in the sea-surface micro-layer removes ozone from the atmosphere (dry deposition) (Ganzeveld et al., 2009) and

results in the emission of inorganic iodine (HOI and I$_2$) into the atmosphere (Carpenter et al., 2013), which can subsequently catalytically destroy ozone (Chameides and Davis, 1980). A number of model studies have discussed the impact of ocean-sourced iodine on atmosphere composition in the context of air quality (Gantt et al., 2017; Sarwar et al., 2016; Sherwen et al., 2017b), climate (Sherwen et al., 2017b; Saiz-Lopez et al., 2012), aerosols (Sherwen et al., 2017a), and stratospheric ozone (Saiz-Lopez et al., 2015). These atmospheric modelling studies have used relatively simple parameterisations for predictions

of sea-surface iodide.

Early parameterisations for sea-surface iodide were based on limited datasets, and used either an observed range of iodide concentrations (Coleman et al., 2010; Chang et al., 2004), or a reported relationship with biogeochemical parameters (e.g. chlorophyll (Oh et al., 2008) or nitrate (Ganzeveld et al., 2009)). However, more recent attempts (Chance et al., 2014; MacDonald et al., 2014) have focused on using correlation analysis to fit compilations of observed iodide concentrations to a

variety of commonly measured sea-surface variables, notably sea-surface temperature, but also chlorophyll, salinity, and nitrate. A summary of parameterisations that have been used in previous studies is given in Appendix Table A1. Compilation of all available observations confirmed a strong latitudinal gradient, and identified sea-surface temperature as the strongest single predictor of iodide concentration (Chance et al., 2014). This approach has led to the equation Eqn. 1 from Chance et al. (2014) and Eqn. 2 from MacDonald et al. (2014).

$$I_{aq}^-(nM) = 0.225 \cdot T(^\circ C)^2 + 19 \tag{1}$$

$$I_{aq}^-(nM) = 1.46 \times 10^6 \cdot \exp(\frac{-9134}{T(^\circ K)}) \cdot 1 \times 10^9 \tag{2}$$



Fig 1 shows the global annual mean distribution of sea-surface iodide calculated using these parameterisations (Eqn 1 and 2) and sea-surface temperature fields (Locarnini et al., 2013). Although both equations predict a similar distribution (higher concentrations in tropical waters and lower in polar waters), Eqn 1 generally predicts iodide concentrations 2-4 times higher than Eqn. 2. In developing Eqn. 1, Chance et al. (2014) compiled iodide observations from both coastal and non coastal sites. However, Eqn. 2 used a relatively small subset (14%) of these observations, which did not include coastal sites, which may explain the lower concentrations. Eqn. 2 also has an Arrhenius form, which could be interpreted to suggest that iodide concentrations are controlled by abiotic reaction kinetics. However, this has not been demonstrated, and Chance et al. (2014) discussed how microbiological activity and oceanic mixing are currently thought to be the primary controls. The choice of different parameterisation (Eqn. 2 versus Eqn. 1) results in a difference of 50% in the calculated global emissions of iodine into the atmosphere (Sherwen et al., 2016a).

Considering the need for spatially-resolved sea-surface iodide field by models and the paucity of observations, parameterisations are required that can yield predictions from ancillary variables. This is a regression problem and a number of approaches are available. Conventional linear and linear multi-variant approaches have been used in the past (e.g. see summary in Appendix Table A1). However, they need to assume a functional relationship between the dependent and independent variables. Another approach is machine learning, which uses algorithms to build predictive models. These algorithms take a different approach and use a non-parametric formulations. Machine learning approaches range from interpretable options such as the "Random Forest" algorithm (Breiman, 2001) to less interpretable ones such as artificial neural networks (Gardner and Dorling, 1998). On the more interpretable end, machine learning algorithms are being used increasingly within environmental sciences, with recent examples including linear Ridge Regression and Random Forest models to replace computationally-expensive processes (Keller and Evans, 2018; Nowack et al., 2018) and Gaussian Process emulation to explore model biases on a global scale (Lee et al., 2011; Revell et al., 2018).

Here, we use a recently expanded compilation of sea-surface iodide observations (Chance et al., 2019b) to build a new sea-surface iodide parameterisation using a data-driven machine learning approach. We choose to use the Random Forest Regressor (RFR) algorithm (Breiman, 2001; Pedregosa et al., 2011), which is relatively simple and produces results that are also easy to understand. We aim to be able to predict global sea-surface iodide based on observations and ancillary physical and chemical variables (e.g. sea-surface temperature, depth, and salinity etc.) from a number of publicly available sources. We first describe the input datasets we use (Sect. 2), then we explain the methodology taken (Sect. 3), and finally present the predictions at observational locations and globally (Sect. 4). We make the resulting high resolution, global, monthly dataset of predicted iodide available to the community (Sherwen et al. (2019); DOI:https://doi.org/10/gfv5v3). When new observations become available, they will be incorporated into the model and updated versions will be provided through a "living data" model.

## 2  Input datasets

Chance et al. (2019b) provides a compilation of the available 1342 sea-surface (< 20 m depth) iodide observations. The dataset is available from the British Oceanographic Data Centre (BODC, Chance et al. (2019a); DOI:https://doi.org/10/czhx).





It includes 45 % more data points, and has greater spatial coverage, than the previous compilation of 925 observations (Chance et al., 2014). Observations are categorised in Chance et al. (2019b) as "coastal" or "non-coastal", according to the designation of their static Longhurst biogeochemical province (Longhurst, 1998). We adopt the same categorisation here. This sea-surface iodide dataset then forms the dependent variable for our regression.

We require a number of physical, chemical and biological parameters as the independent variables in our regression models. Consistent in-situ measurement of these parameters are not available for the iodide observations. Thus we have used a number of ancillary datasets (Table 1) to provide this information. There are a number of criteria for these datasets: they need to be available at an appropriate resolution as a gridded product; they need to represent potential processes that could control iodide concentrations and they need to be in some way orthogonal to the other independent variables. Gridded datasets of dissolved
organic carbon (e.g. Roshan and DeVries (2017)) and phytoplankton primary productivity (e.g. Behrenfeld and Falkowski (1997)) may have some usefulness, but they themselves are built using statistical models with other variables and thus we do not use those here. The selected ancillary variables (Table 1) were first extracted from their native resolution using the nearest-neighbour method, onto a consistent high-resolution monthly grid ($0.125°\times0.125°$, $\sim12.5$km). This horizontal resolution was used as this is the highest resolution of the current generation of global atmospheric chemistry simulations (Hu et al., 2018). We
calculate monthly means because the chemical lifetime of iodide in the surface oceans is thought to be at least several months (Campos et al., 1996; Žic et al., 2013), and possibly years (Edwards and Truesdale, 1997; Tsunogai Shizuo and Henmi, 1971). Indeed, the lifetime of iodide is thought to be sufficiently long that, where deep vertical mixing occurs on a seasonal timescale, this may be the dominant loss process from surface waters (e.g. Chance et al. (2010)). The values for bathymetric ocean depth we set to a minimum depth of 2 metres, to remove terrestrial locations, and the same value was used for all months.

For each iodide observation, the nearest point in space and time was extracted from the high resolution gridded ancillary data. For the 31 iodide observations where a month was not available (Luther and Cole, 1988; Tsunogai Shizuo and Henmi, 1971; Wong and Cheng, 1998), an arbitrary month was chosen (of March for Northern hemispheric observations and September for Southern hemispheric observations). Outliers within the observations are removed as described in Sect. 2. A further single dataset (Truesdale et al., 2003) was also excluded from this analysis. This is discussed in Appendix Sect. A1.

## 25  3  Methods

Here we first explain the way in which we use the machine learning algorithm (Sect. 3.1). We then explain how we have calculated uncertainty (Sect. 3.2), how observations considered outliers have been removed from the data (Sect. 3.3), and how we have decided which ancillary variables (e.g. temperature, salinity, etc) to use as independent variables for an ensemble prediction (Sect. 3.4). Finally we describe the interpretable ensemble prediction model that results from this methodology in
both numerical and graphical terms (Sect. 3.5).



### 3.1 Random Forest Regressor algorithm

As the aim here is to predict a continuous numerical value for sea-surface iodide, a regression approach is taken. As discussed in the introduction, previous approaches have been made to parameterise sea-surface iodide, and the most commonly used relationships employ sea-surface temperature as the predictor variable. Here we take a different multivariate and non-parametric

approach, using the computationally cheap and interpretable Random Forest Regressor (RFR) algorithm (Breiman, 2001; Pedregosa et al., 2011).

Random forest regression is based on finding a number of decisions trees, which predict the dependent variable. As all of the trees contribute to the prediction and they are collectively referred to as a "forest". These trees can be explained as a record of the way the algorithm has linearly traversed a subset of the training data, splitting the data into two parts at each decision

point or "node" in a way that minimised the internal differences of the parts. The best split is chosen between the available variables based on an error metric (e.g. mean square error) and this process is continued until a criterion of purity is reached or a minimum number of data points are left from a split. This is essentially a classification problem. The prediction of the forest is the mean value of the prediction of all of the different decision trees, which attempts to make the results more of a regression problem. More details of this approach can be found in Friedman et al. (2009).

This approach differs to previous approaches which have individually tested proposed relationships and selecting the best performing model(s) as a parameterisation (e.g. Table A1). Here, an algorithm uses the data it is provided to build a model that gives a prediction and therefore it is the data itself that defines the model that is used to predict new values. A key difference of this approach is also that only a subset, the "training" set, is used to build the model and the rest (or "withheld" set) is then used to test the performance of the model. Here we use 80 % of the data for the "training" set and use the remaining 20 % as

the "withheld" set (also commonly referred to as the "testing set").

To ensure that the models built are generalisable and mitigate overfitting, the Random Forest approach used here artificially increases the randomness within the forest (Pedregosa et al., 2011). This is done by randomly combining single decision trees by an approach referred to as "bootstrap aggregation" or "bagging" (Breiman, 2001; Tong et al., 2003). This additional "bagging" approach randomly samples observations within the training dataset and so mitigates over-fitting of the trees to the

dataset (Friedman et al., 2009). Furthermore, to maintain the statistical distribution between the training and withheld datasets and the dataset as a whole, a stratified sampling approach is used to randomly select data within the quartiles of the dataset.

Machine learning algorithms can generally be tuned to increase performance using settings called hyperparameters. However, Random Forests are known to generally perform well without tuning. The default hyperparameters therefore were used here (Pedregosa et al., 2011), except for increasing the number of trees ("n_estimatators") from 10 to 500. Mean Square Error

(MSE) was used as the criterion for evaluating each split (also referred to as a "node"). The maximum number of "features" (the ancillary variables provided to the algorithm, such as temperature or nitrate concentration) considered when looking for the best split is set to the number provided to the algorithm. The number of splits a tree is allowed to make ("max_depth") is not restricted and further nodes are made until leaves contain less than two samples ("min_samples_split") and a minimum of one ("min_samples_leaf"). All forests are built here use bootstrapping.





## 3.2 Error and uncertainty calculations

Understanding the errors and uncertainties in the global iodide distribution is important due to any sensitivities to this value within the modelled Earth system. We consider three sources of error in our predictions: the "dataset selection" error due to the splitting of the dataset into training and withheld parts; the "model selection error" due to the choice of dependent variables;

and the "observational error" on the iodide measurements.

To quantify the "dataset selection" error, we construct models from 20 pseudo-random splits of the dataset into training and withheld parts. The hyperparameters and input ancillary variables are kept the same for the generation of the 20 models, so that the only difference between the models is the training dataset. These 20 models are then used to predict the withheld data. Performance metrics (Root Mean Square Error (RMSE) and average absolute prediction etc.) can then be calculated for

each model. This gives a range of 20 values, which can then be converted to a percentage range as the error. This is done by dividing the maximum within the range over the maximum value to give a maximum value and minimum within the range over the maximum value to give minimum value. Significant differences between model's performance metrics would suggest important sensitivity to the training / withheld dataset splits.

We define the "model selection" error as the uncertainty resulting from the choice of input ancillary variables. A number

of combination of input variables are possible in generating the models, and each will generate a different prediction. We quantify this error as the difference in performance against the withheld dataset and prediction value (e.g. average global value). Similarly to our calculation of "dataset selection" error, this can be converted to percentage error by considering the range in these values and dividing them by minimum and maximum values.

For the "observational error" we refer to Chance et al. (2019b), who provide individual error estimates for each of the iodide

observations in the data compilation. Over half (51 %) of the data points have an error of 5 % or less, and a further ∼25% have an uncertainty in the range of 5-10 %.We therefore use a value of 10 % as a conservative estimate of the "observational error".

## 3.3 Outlier identification and removal

Our dataset consists of values for ancillary variables and iodide concentration for all of the 1342 measurement locations in the observational dataset (Sect 2). As discussed in Sect 3.1, we split this dataset into two parts: (i) a training set for use in building

and optimising models, and (ii) a withheld set to evaluate the models built. Particular care was taken to ensure the withheld and training datasets were representative of the entire dataset in the way the models built, therefore improving performance and "generalisability" to unseen data (See Sect. 3.1).

We take a Random Forest Regressor (RFR) model built with variables that were intuitively assumed to give a reasonable ability to differentiate the observations (using depth, temperature, and salinity as the independant variables - abbreviated to

"RFR(DEPTH+TEMP+SAL)" following Table 1). The "RFR(DEPTH+TEMP+SAL)" model was then used to explore the variation of error in the predictions using the "dataset selection" error approach described in Sect. 3.2. This builds multiple versions of the same model with different splits of training data and yields a distribution of Root Mean Square Error (RMSE)



in the predicted iodide for withheld data as summarised in the final column of Table 2 and shown graphically in Appendix Fig. A1.

We define outliers here as values greater than the $3^{rd}$ quartile plus 1.5 times the interquartile range (Frigge et al., 1989). Removing these forty nine values categorised as outliers ($>$309.5 nM) leads to a vast improvement in the RMSE error in the
ensemble prediction from 95.1 nM to 37.6 nM (Table 2). This is shown graphically in Appendix Fig A1, with the other subsets of the data explored (Table 2). This demonstrates that the high values are not well enough represented by the dataset to be able to be captured by the RFR approach. The removal of these high values from the dataset can also be justified as the driver for these concentrations is not yet well understood (Chance et al., 2014, 2019c; Cutter et al., 2018).

Removing these outliers reduces RMSE in the prediction with the twenty independent model builds from 48.2 nM to 2.3 nM
($3^{rd}$ quartile - $1^{st}$ quartile). Once these outliers are excluded, more modest changes in average RMSE are then seen if models are built only using coastal or non-coastal data. Fig. A1 also shows this is seen when removing lower salinity data ('Salinity $\geq$30 PSU & no outliers'), which is indicative of estuarine water. This highlights the strength in this approach's ability to predict iodide in different biogeochemical regions (i.e. not just coastal or non-coastal locations).

An additional removal of a single dataset of nineteen observations from the Skagerrak strait (Truesdale et al., 2003) was
made due to it exerting a disproportionate influence on iodide prediction in high Northern latitudes ($>$=65 °N), an area that is almost entirely unconstrained by local observations. We note that the Skagerrak is relatively unusual oceanographically, being an estuarine location with high ship traffic, and is considered unlikely to be an analogue for iodine speciation in the Arctic. This is decision is discussed further in Appendix A1 and the predictions made including this dataset are also included in the shared output (Sect. 5).

From here, only the 1293 observational points excluding outliers and the data from the Skagerrak strait (Truesdale et al., 2003) are used.

## 3.4   Selection of ancillary variables and building an ensemble prediction

To decide which ancillary variables (temperature, salinity, etc, - see Table 1 and Sect. 2) should be used to predict sea-surface iodide concentration, RFR models were built and evaluated with different combinations of variables. Thirty eight combinations
were considered (see $1^{st}$ column of Appendix Table A2) .

The top twenty performing models, based on their Root Mean Square Error (RMSE) against the withheld data, are plotted in Fig. 2, alongside existing parameterisations. The standard deviation for all predicted values is also shown to illustrate variation in the predictions. A complete list of the performance and of all models built here and their performance is given in the appendix (Table. A2).

The RMSE values in Fig. 2 shows the increased skill present in the new predictions compared to the existing parameterisations. The RMSE improves from the 75.3 and 50.2 nM found for the Chance et al. (2014) and MacDonald et al. (2014) parameterisations, respectively, to 33.2-37.4 nM for the top ten models created here. Only modest gains are seen in RMSE between models with three variables or more.





The best-performing model in the list is only marginally better than the $10^{th}$ best performing, therefore there is not an obvious "best" performing set of ancillary variables. Thus going forwards we use an ensemble prediction approach based on the mean value from an ensemble of the 10 top-performing models.

### 3.5 Model Descriptions

Unlike many machine learning approaches, the Random Forest Regressor algorithm is interpretable. The decision trees can be visualised to explain the main features driving the splits. Figure 3 shows schematically the whole regression approach taken here. Panel (a) shows single trees, of which 500 are built with the same input variables and then combined into forest (b). Then this forest is combined with the nine other top-performing models (made from different combinations of ancillary variables) to make an ensemble (c). The ten predictions of (c) are then arithmetically averaged into a single prediction, which thus includes

the predictions of 5000 trees with 10 different combinations of input variables. In Fig 3a, the colour of a limb or "branch" following a node is given by the variable driving that split within the training dataset. For Fig. 3b and 3c it shows the percent of that times that a variable drives that node within the forest. The value of the ancillary variable that sets the split is shown inside the circle (a,b,c). The thickness of the branch scales to the throughput of training dataset samples contained within that split. The trees are shown to a depth of five nodes for aesthetic reasons and due to increased divergence of the trees within a

forest the deeper you go. However the trees themselves are unlimited in the depth they can reach.

The first and larger splits in the data at decision "nodes" in the models can be simply read, which can provide understanding of the main variables driving the initial and largest splits in the prediction. For all models in the ensemble, the initial split is driven by temperature, with a split occurring at around 21.1 °C (with a standard deviation of 1.2 °C). The data is then split by two further nodes from this, a left and right hand split (e.g. Fig. 3b). If depth or temperature is present as a variable, then they

drive the majority of the next splits. If depth is not present as a variable, then either nitrate or mixed layer depth (MLD) is the most common variable to dictate the split in the data at the next node in the tree. Thus a qualitative way of interpreting the initial splits of the dataset would be to say that the model is primarily differentiating between warmer and shallower locations.

## 4 Results

Here we evaluate the performance of the ensemble prediction against the observational dataset (Sect. 4.1) and then we explore

the predicted global monthly surface concentrations (Sect. 4.2).

### 4.1 Prediction of iodide at observational locations

Figure 4 shows a point-by-point comparison between parameterised and observed iodide for the: entire dataset; the withheld dataset; withheld coastal dataset and withheld non-coastal dataset. Predictions are shown for the ensemble Random Forest Regressor (RFR) approach described here, and for both the Chance et al. (2014) and MacDonald et al. (2014) parameterisations.

The Root Mean Square Errors (RMSE) of observed and predicted values are given in the Figure 4 and in Table 3.





The new ensemble prediction is the best performing, with a lower RMSE (35 nM) compared to the existing parameterisations (75 nM and 50 nM for the MacDonald et al. (2014) and Chance et al. (2014), respectively) for the withheld data. Both the new parameterisation and Chance et al. (2014) parameterisation are relatively unbiased (both have best fit line slopes of 0.84, against the withheld data), but the new parameterisation shows less noise than Chance et al. (2014). MacDonald et al. (2014) shows a significant low bias and significant noise. The improved skill from the RFR ensemble is consistent for both coastal and non-coastal observations.

Figure 5 shows comparisons between the probability distribution functions (PDFs) of the observed iodide and the predictions, together with the PDFs of the biases for the entire, coastal and non-coastal withheld datasets. The PDF of the new parameterisation shows the greatest similarity to the observations. The PDF from Chance et al. (2014) show a similar range to the observations and structure to the observations, whereas the PDF from MacDonald et al. (2014) shows again a significant underestimate. The bias plots show the new predictions are generally clustered around zero with a relatively narrow peak. Chance et al. (2014) is again roughly clustered around zero but shows a wider peak. The largest biases are found from MacDonald et al. (2014) which systematically underestimates observed iodide concentrations.

The "dataset selection" error, which shows the the influence of the choice of how the dataset is split into training and with data on model prediction, is described in Section 3.2. Within the 20 member ensemble of different testing/withdrawn choices, the average variation in RMSE was 8.4 nM (5.9-11.02 nM) and in the range of average predicted values was 6.1 nM (5.4-6.6 nM). This translates to a percentage error of 16.1-29.5 % on the RMSE and 5.6-7.0 % on the average predicted value.

The "model selection" error, which is the influence of the different independent variables used, is described in Section 3.2. The difference in the average prediction of the 10 members of the ensemble is 1.8 nM (with a range of average prediction from 96.0 to 97.8 nM) and the range of the difference in model performance is 3.9 nM (33.2-37.2 nM). As a percentage this "model selection" translates to a percentage uncertainty on the the RMSE of 10.6-11.9 % and on the average of 1.8-1.9 %.

The "dataset selection" and "model selection" compares to an error on the observations of ∼10 %. Uncertainty from "dataset selection" has a far greater effect on the prediction error than "model selection". This is can be expected due to the small dataset size. The combined error in the prediction ("dataset selection"+"model selection" error) is either comparable to (7.4-8.9 % in terms of average prediction) or greater (27-41 % in terms of RMSE) than the observational error.

From this analysis we have shown that the new ensemble RFR model performs significantly better than those currently in the literature. We now turn to explore the predicted global distribution of sea-surface iodide using our ensemble model.

## 4.2 Global sea-surface iodide distribution

From the ensemble prediction system we calculate monthly global grids ($0.125° \times 0.125°$, ∼12.5 km) of sea-surface iodide using the gridded ancillary data (Sect. 2). The annual average spatial predictions are shown in Fig. 6 with the observations overlaid in circles. Similar to previous work, annual average maximum concentrations of 220 nM are found in tropical and coastal regions (e.g. Oceania and in the Caribbean/Gulf of Mexico) with the lowest concentrations in mid-latitude waters (22.4 nM). Seasonal variability is also seen within the monthly prediction (Appendix Fig. A3). However, this spatial and temporal variability is bot well constrained by observations. For example, the some of the highest concentrations are predicted for the



South China sea, a region without any observations (Fig. 6). Some features are visible in the concentration field appear to be associated with deep bathymetric features (e.g. the higher concentrations over the mid Atlantic ridge - Fig. 6)), even though a physical explanation for such a link seems unlikely.

Summary statistics on the global predictions are shown in Table 4. These show that, as for comparisons at the observed
locations (Section 4.1), the ensemble prediction is broadly in between the two existing parameters. The new ensemble model predicts a mean value of 106 nM (with members ranging from 102.3 to 108.8 nM), with predicted values from existing parameterisations ranging from 58.9 (MacDonald et al., 2014) to 128.1 nM (Chance et al., 2014).

The annual latitudinal average of these fields, together with predictions from Chance et al. (2014) and MacDonald et al. (2014), and the observations are shown in Fig. 7. Far greater structure is seen compared to the two existing parameterisations
(Fig. 7) due to the multivariate and non-parametric ensemble approach used here. All parameterisations capture the broad observed feature of decreasing iodide from lower to higher latitude. The new predicted values lay between Chance et al. (2014) and MacDonald et al. (2014) in the tropics, however, within the polar regions, the new prediction is significantly higher than both of the previous parameterisations. The lower concentrations in the predicted values from MacDonald et al. (2014) for most of the global sea-surface is clear.

The "dataset error" is found for the 20 models with different training data splits, as described in Sect. 3.2. This gives an uncertainty in the form of a average range in predicted global mean surface iodide for all of the multiple build of ensemble members of 4.0 nM (2.8 - 5.0) compared to a annual mean prediction of 106 nM. This maximum and minimum of this range in predicted values can then be divided by the minimum and maximum predicted global mean surface iodide values (98 nM and 109.3 nM, respectively) to give percent range of 2.6 to 5.0 %. This is lower than that calculated for the individual locations
of observations (Sect. 4.1) due to large global areas being similar in chemical and physical regimes compared to the subset of sampled locations within the observations.

The "model selection" error due to variability within ensembles' 10 members, generated with different independent variables, gives a global average surface concentration between 102.3 to 108.8 nM. This range in prediction gives a "model selection" error of 6.45 nM, which equates to 6.0-6.3 %. Like with the global uncertainty from "dataset selection", the global
value would be expected to be lower than the uncertainty at the specific locations of the observations (Sect. 4.1) due to the more homogeneous nature of the predicted areas. However, a greater variation is seen from different model predictions than within predictions for the observation locations. This highlights the importance of the different ancillary variables considered here and also therefore the strength gained from the ensemble approach taken here.

Within members of the ensemble, variation is modest except for two ensemble members which divergence north of $>=65\,^{\circ}$N
(Appendix Fig A2). As noted earlier (Sect 2), the values in this region are very poorly constrained by the observational dataset (Fig 6).

In addition to the three errors we described above, we also attempt to gain to an understanding of the spatial uncertainty in the ensemble prediction. We do this via calculating the differences in the predicted spatial fields from the 10 ensemble members. Fig. 8 shows the monthly average of the standard deviation of the 10 model ensemble as a percentage of the annual mean of
35 the ensemble prediction. This is also shown in absolute terms in the Appendix (Fig A4). Uncertainties are largest at the poles



where predicted concentrations are lowest and where (at least in the Northern hemisphere) very few observations are available to constrain the system. The Southern Oceans show an distinct pattern, where values close to coastal Antarctica appear well constrained but values further north appear poorly constrained.

## 5 Data availability

The monthly ensemble mean and standard deviation between ensemble members for the main prediction presented here ("RFR(Ensemble)"), along with the individual ensemble members are archived at the United Kingdom's Centre for Environmental Data Analysis (CEDA) as monthly files in NetCDF-4 format (Sherwen et al. (2019); DOI:https://doi.org/10/gfv5v3). To enable use in atmospheric and oceanic models, we have additionally bi-linearly re-gridded the outputted fields onto common model grids (Appendix Table A5) using the open-source Python xESMF package (Zhuang, 2018). We recommended use of the standard output provided, but have also provided the predictions made by the model with the Skagerrak dataset (Truesdale et al., 2003) included (which was excluded from the analysis presented here, as discussed further in Appendix Sect A1).

Ancillary data extracted for observation locations and used to predict spatial fields is available from sources stated in Table 1. Iodide observations are described by Chance et al. (2019b) and made available by the British Oceanographic Data Centre (BODC, Chance et al. (2019a); DOI:https://doi.org/10/czhx).

## 6 Discussions and conclusions

Here we have explored the ability of an algorithmic approach combined with various physical and chemical variables to predict sea-surface iodide, without aiming to represent the biogeochemical or abiotic processes occurring. This approach instead gives a data-driven "best guess" at concentrations and an ability to quantify where the greatest uncertainty lies. However, certain features such as prediction of an apparent relationship between ocean bathymetry and sea-surface iodide concentrations, where the ocean is very deep (e.g. over the Mid-Atlantic Ridge) are unlikely to have a plausible physical explanation (Fig 6).

The new spatial prediction presented here differs from what has been used previously in atmospheric models (e.g. Chance et al. (2014); MacDonald et al. (2014)). Although the average value lies between these parameterisations, the prediction is closest to that from Chance et al. (2014) even with larger values found at higher latitudes. As most atmospheric models have used the iodide parameterisation from MacDonald et al. (2014) (Appendix Table A1) to calculate ocean iodine emissions, a higher emission would therefore now be expected. This would result in larger decreases in tropospheric ozone burden than previously suggested (Sherwen et al., 2016a). A higher iodide sea-surface concentration would also result in a greater calculated ozone deposition (Luhar et al., 2017; Sarwar et al., 2016).

We have calculated the errors in sea-surface iodide concentrations at observational locations due to the "dataset selection" of 16.1-29.5 %, and due to "model selection" of 1.8-1.9 % (Sect 4.1 and 4.2). These error estimates can be compared to an approximated error in the observations of ∼10 % (Chance et al., 2019b). Considering the average predicted concentration globally here is 106 nM (Sect. 4.2), these errors are notable. The greatest driver in error is the "dataset selection". More





observations, and particularly observations representative of under-sampled areas and seasons, will be required to reduce this error. The error caused by "dataset selection" is also reduced when the predictions are considered spatially over the global sea-surface.

The choice of the algorithm used here is subjective and numerous other options are available. The Random Forest Regressor was chosen due to appropriateness for the continuous regression task performed here, its relatively cheap computation cost, and its interpretability. Considering the greatest uncertainty is driven by the paucity and sparsity of observations, using more complex techniques would not be expected to yield particularly different or drastically better results, considering other trade-offs.

We have developed a new way to build a spatially and temporally resolved dataset from a spatially and temporally sparse input of observations. This has allowed for use of use more of the observations, than traditional approaches, which is particularly important with a paucity of data. This approach has demonstrated a large improvement in skill in terms of capturing observations compared to the existing parameterisations in use. It captures the increasing trend of iodide with latitude seen in the observations, as well as the greater spatial variation seen in the observations.

## 7 Code availability

Data analysis and processing used open-source Python packages, including Pandas (Wes McKinney, 2010), Xarray (Hoyer and Hamman, 2017) and Scikit-learn (Pedregosa et al., 2011). Spatial re-gridding used the xESMF package (Zhuang, 2018). Plots presented here were created using the Matplotlib (Hunter, 2007) and Seaborn (Waskom et al., 2017) python packages. The decision tree figures (Figs. 3 and A5) were made using the TreeSurgeon (Ellis and Sherwen, 2019) package.





## Appendix A

### A1    Removed Skagerrak dataset

Ideally with sparse datasets, as much data as possible would be included for training the regression models used. If a feature in the data is different enough to the rest of the dataset and not sufficiently be represented for the regressor model to characterise

it, then it has potential to introduce a large "dataset error" (See Sect. 3.2 for details). This was shown when the iodide values above the outlier threshold were included (Sect. 3.2). There could be many other affects of including data that is significantly different to the rest of the dataset.

The data from the Skagerrak strait (Truesdale et al., 2003), which is included in the (Chance et al., 2019b) compilation of iodide data, was excluded from this analysis. This is because upon inclusion, high iodide at high latitudes ($>= 65\ °N$) are

calcuated (Appendix Fig. A6). An increasing trend is seen with latitude, reaching values comparable to the highest predicted values in the tropics. This region has a paucity of observations within the Chance et al. (2019b) compilation and there are none further north than Iceland. This means that any prediction in this region would be unconstrained by observations. Exclusion of this data leads to Fig 7 where iodide is generally constant above $65\ °N$.

The Skagerrak strait data (Truesdale et al., 2003) is also from region where the observed ancillary variables compare poorly

with those extracted from ancillary datasets. Observed salinity is between 24.0 and 33.5 PSU, whereas the climatological value is 31.7 to 35.8 PSU. This equates to a bias of the climatology versus the in-situ observations of up to 9.6 PSU or 40 %. The Skagerrak is biogeochemically different from the Arctic, and its large influence on predicted values in the Arctic may arise simply from its latitudinal proximity, given the lack of observations from the regions itself.

The area this dataset is sampling in is also unusual in the Chance et al. (2019b) compilation due to its estuarine nature.

However, this cannot entirely explain its behaviour as their are other estuarine datasets included (such as those from around the Chesapeake Bay (Luther and Cole, 1988; Wong and Cheng, 1998, 2008)) which do not cause the same issue.

As the feature of high predicted Arctic iodide is driven by a single dataset of 19 samples (of which 4 would be removed as outliers) from a different region, it is highly uncertain. Not only do the in-situ salinity observations compare poorly to the extracted ancillary ones, but the location itself represents a heterogeneity within the Chance et al. (2019b) compilation as it has

relatively high observed iodide concentrations. It was therefore omitted from the analysis presented within this paper. However results with this dataset are included in the shared data outputs. It is hoped further observations $>65\ °N$ could offer more insight into this uncertain region and also on to the observations in the Skagerrak strait.





*Acknowledgements.*  We all thank those who have provided their published and unpublished data to the Chance et al. (2019b) dataset and the British Oceanographic Data Centre (BODC) for hosting this, as well as all support staff involved.

We gratefully acknowledge those who have worked to compile and make avialible the ancillary datasets used here (Table 1). These include the World Ocean Atlas (WOA), Ocean Biology Processing Group (OBPG) at NASA's Goddard Space Flight Center, and the General Bathymetric Chart of the Oceans (GEBCO) teams.

We thank Tim Jickells, Peter Liss, David Stevens, and Martin Wadley for their useful conversations on and comments about this work.

We gratefully acknowledge funding from Natural Environment Research Council (NERC) through grants "Iodide in the ocean: distribution and impact on iodine flux and ozone loss" (NE/N009983/1) and "Big data for atmospheric chemistry and composition: Understanding the Science (BACCHUS)" (NE/N009983/1).





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



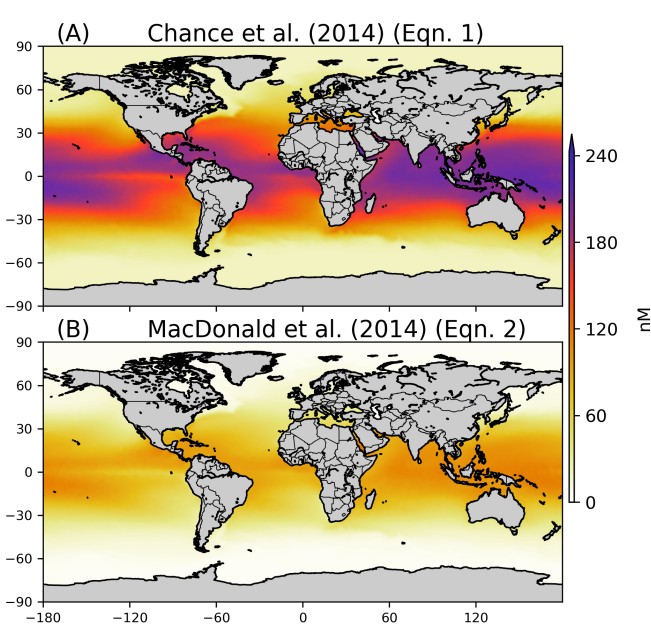

**Figure 1.** Annual average sea-surface iodide concentrations predicted by (A) Eqn. 1 from Chance et al. (2014) and (B) Eqn. 2 from Mac-Donald et al. (2014). Temperature fields used to make spatial predictions were from the World Ocean Atlas (Locarnini et al., 2013).



**Figure 2.** Random Forest Regression (RFR) model performance (Root Mean Square Error (RMSE), blue) against the withheld data for the top 20 models on left hand y-axis, along with values from the parameterisations from Chance et al. (2014) and MacDonald et al. (2014). Right hand y-axis is standard deviation of the prediction for the withheld data (orange). Top ten performing models and the two exiting parameterisations considered here (Chance et al., 2014; MacDonald et al., 2014) are shown in bold. Parameterisations are ordered by their RMSE. Abbreviations are given in Table 1.



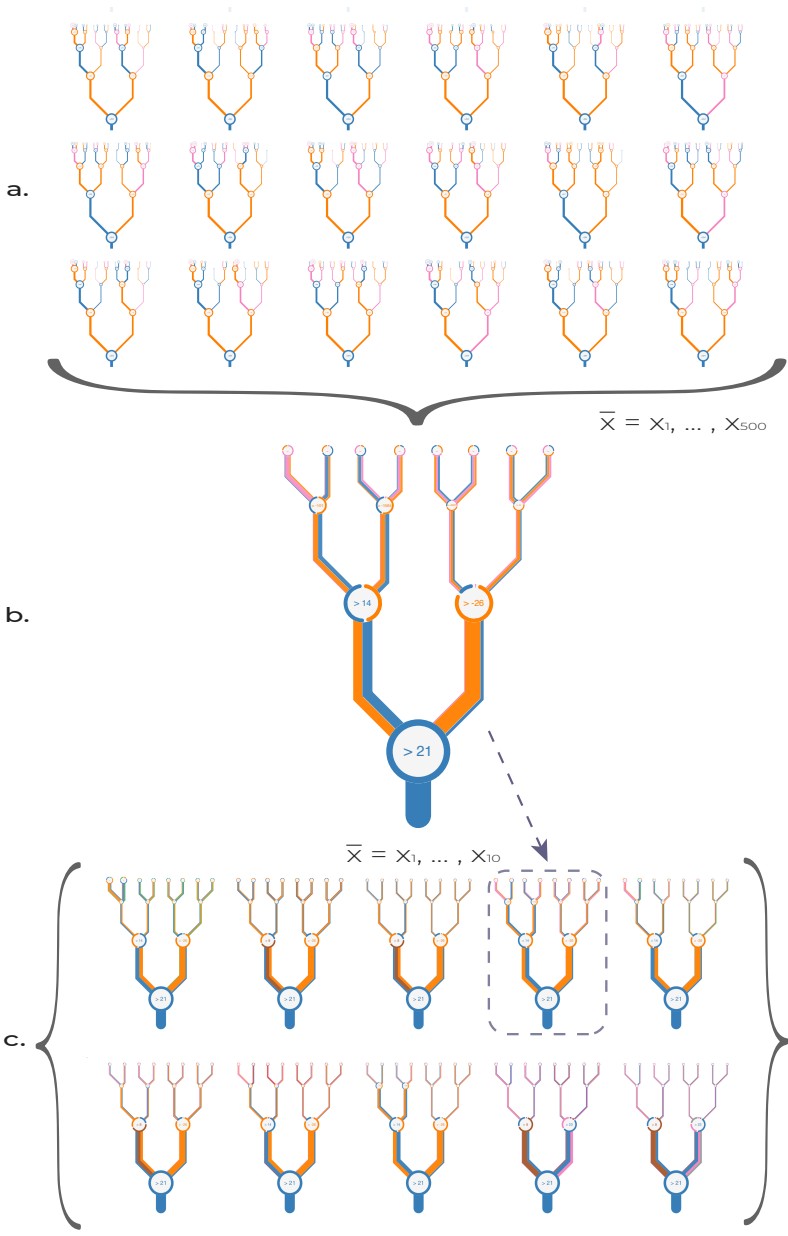

**Figure 3.** Schematic illustration of how (a) multiple decision trees are combined into (b) a forest and then combined into an (c) ensemble. (a) shows individual trees in a forest. (b) represent a forest of 500 trees as a single figurative tree. (c) shows the ten forsts of 500 trees combined into a single prediction. The branches in plots (a)-(c) are coloured by the percentage of the decisions at a given node that are driven by a given variable. That value within the circle gives the value of the main ancillary variable driving a split. Thickness of branches gives the throughput of the dataset through a given node for single trees (a), or the average for plots of forests (b,c). The 10 forests shown as thumbnails in panel (c) are also shown in larger form in Appendix Fig. A5. Variable names are coloured as per the following coloured text: temperature (blue, °C), depth (orange, metres), chlorophyll-A (green, mg m$^{-3}$), salinity (pink,PSU), nitrate (brown, $\mu$g m$^{-3}$), mixed layer depth (MLD; purple, metres), phosphate (red, $\mu$g m$^{-3}$), and shortwave radiation (grey, Wm$^{-2}$).





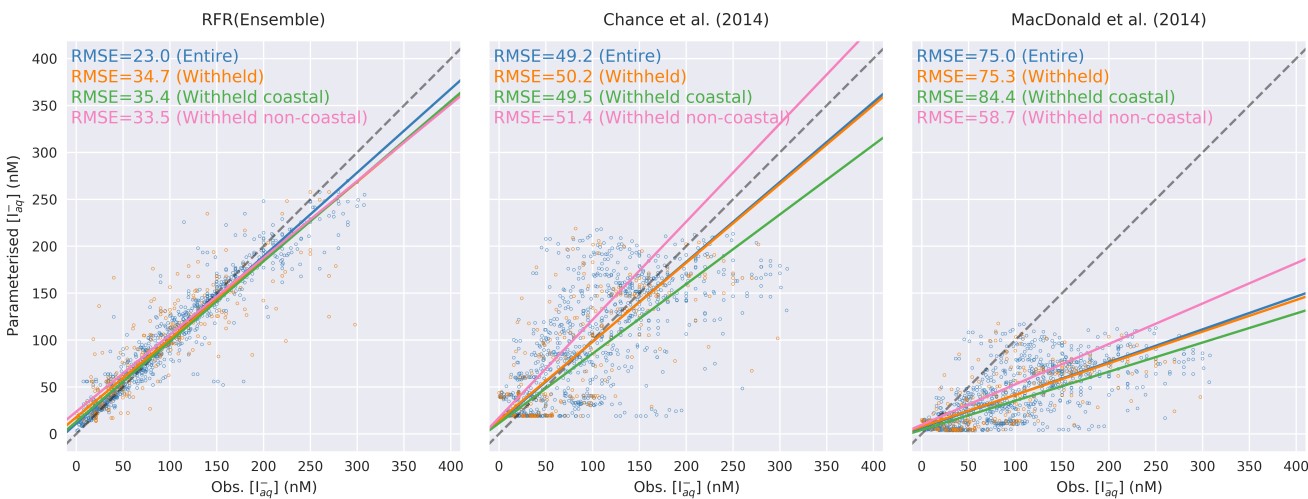

**Figure 4.** Regression plots showing comparisons between predicted values and observations in the entire (blue, N=1293) and withheld data (orange, N=259), withheld data classed as coastal (green, N=157), and the withheld data classed as non-coastal (pink, N=102). Solid lines give orthogonal distance regression line of best fit. The dashed grey line gives the 1:1 line. Root Mean Square Error (RMSE) for each line is annotated by subplot in nM.





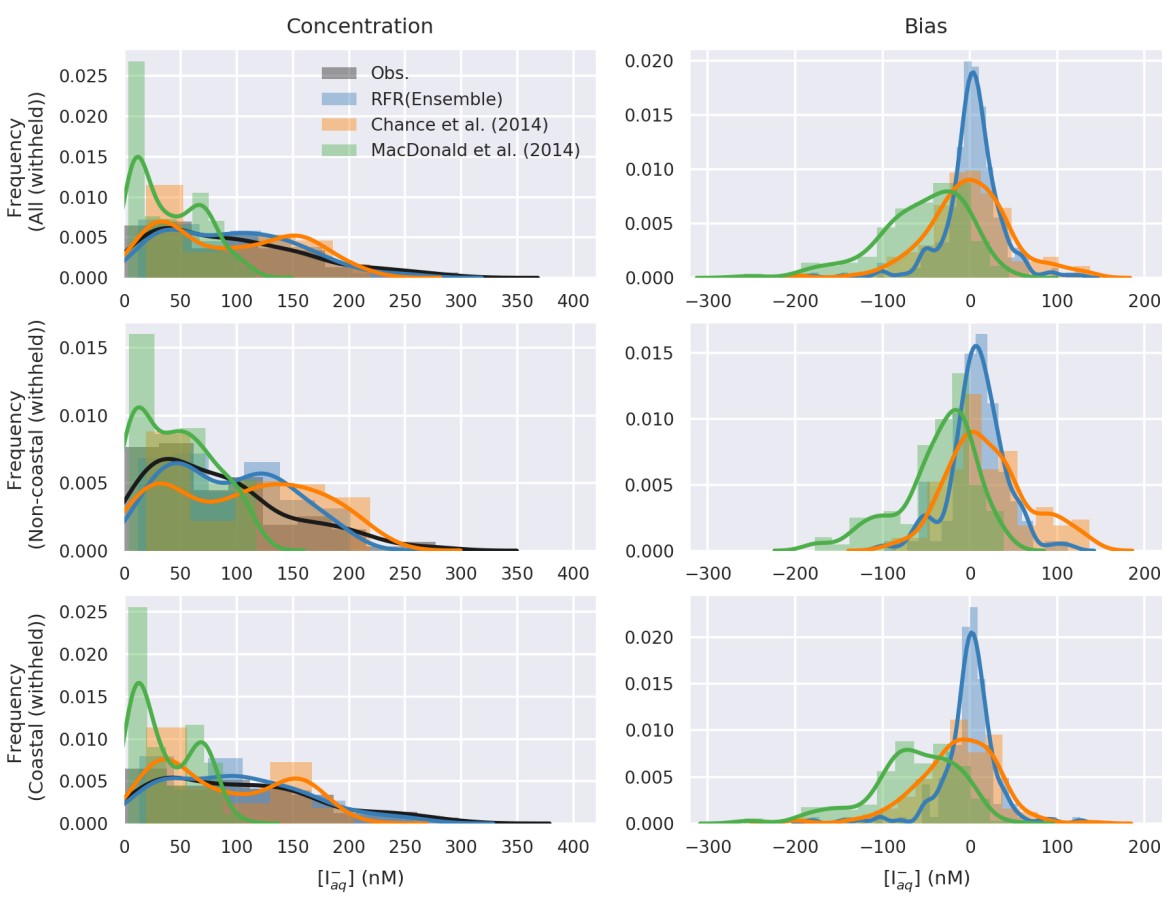

**Figure 5.** Probability density function (bars) and Gaussian kernel density (lines) estimate of observations and predicted concentrations (left), and bias (right, model-observations) in entire withheld dataset (upper, N=259), the withheld coastal dataset (middle, N=157), and the withheld non-coastal dataset (lower, N=102).





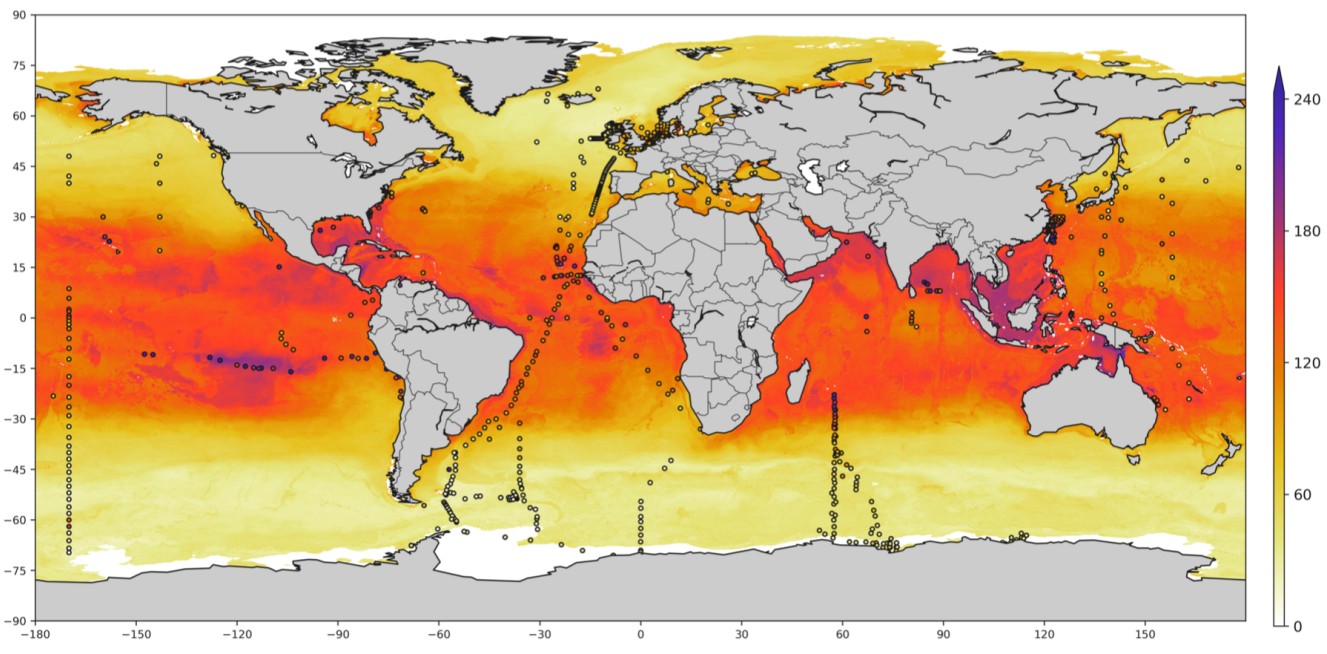

**Figure 6.** Annual average predicted sea-surface iodide for the ensemble of models ("RFR(Ensemble)"), overlaid with iodide observations from Chance et al. (2019b) without outliers. Outliers are defined here as values greater than 3rd quartile plus 1.5 times the interquartile range (Frigge et al., 1989). Only values where that are entirely water are included in spatial the average.



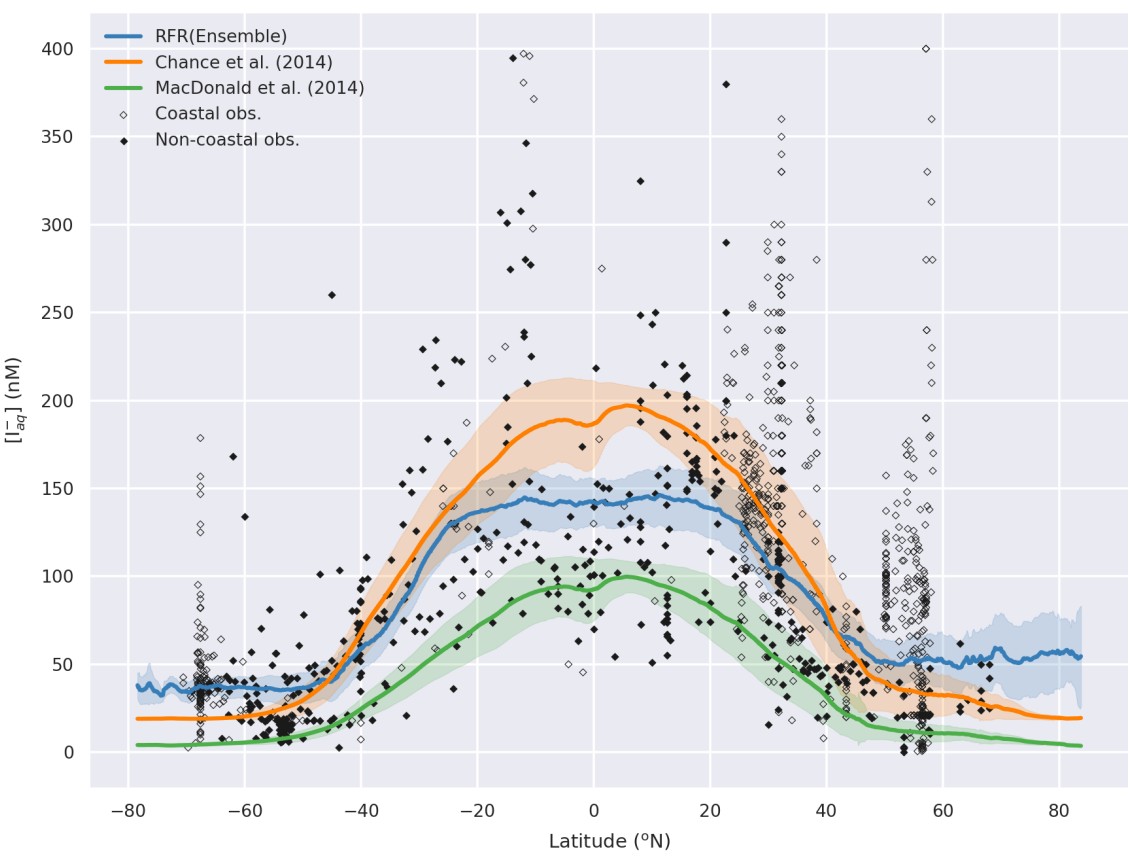

**Figure 7.** Predicted annual average sea-surface iodide plotted against latitude (lines), overlaid with observed concentrations (diamonds). Solid lines give mean values and shaded regions give ($\pm$) the average standard deviation. The standard deviation is the monthly standard deviation across a latitude between all ensemble members ("RFR(Ensmeble)") or within a single prediction for existing parameterisations (Chance et al., 2014; MacDonald et al., 2014). Filled diamonds show non-coastal observations and filled ones show coastal values. Extent of x axis is shown for grid-boxes that are entirely water.





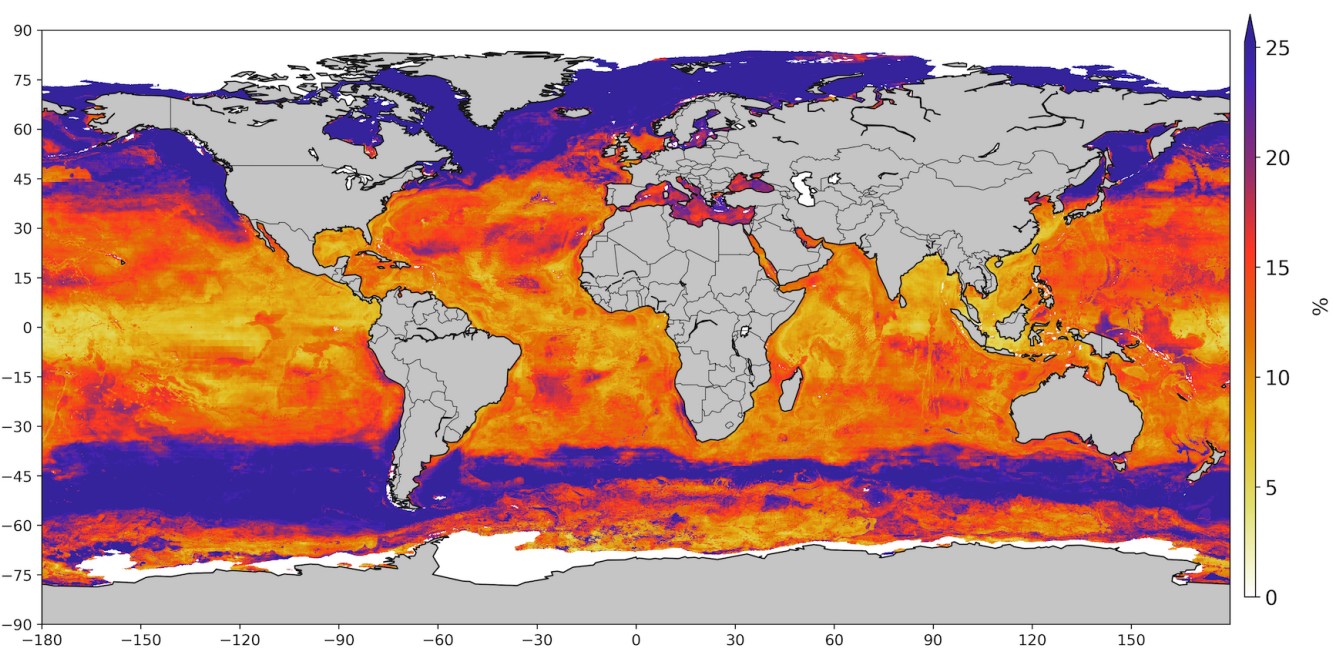

**Figure 8.** Annual average spatial percent uncertainty in predicted sea-surface iodide for the ensemble of models. Percent spatial uncertainty was calculated as the standard deviation in monthly average values for all models, divided by the annual mean. Values are limited to 25 % for contrast, but the maximum plotted value is 77 % in the Northern high-latitudes. Only locations that are entirely water are included in the spatial average.



**Table 1.** Ancillary variables extracted onto a global 0.125°x0.125° (~12.5 km) grid on a monthly basis.

| Field | Abbreviation | Resolution (space, time) | Reference |
|---|---|---|---|
| Sea-surface temperature | TEMP | 0.25°×0.25°, monthly | WOA, (Locarnini et al., 2013) |
| Salinity | SAL | 0.25°×0.25°, monthly | WOA, (Zweng et al., 2013) |
| Dissolved oxygen | O2 | 1°×1°, monthly | WOA, (Garcia et al., 2010) |
| Bathymetric ocean depth | DEPTH | 9 km ( 0.08°), N/A | GEBCO; (Becker et al., 2009; Smith and Sandwell, 1997) |
| Nitrate | NO3 | 1°×1°, monthly | WOA, (Garcia et al., 2014) |
| Phosphate | Phos | 1°×1°, monthly | WOA, (Garcia et al., 2014) |
| Silicate | SIL | 1°×1°, monthly | WOA, (Garcia et al., 2014) |
| Chlorophyll | ChlrA | 9 km, monthly | SeaWIFS, (OBPG, 2014) |
| Mixed layer depth | MLD* | 1°×1°, monthly | WOA, Monterey, G. and Levitus (1997) |
| Shortwave radiation | SWrad | 1.9°×1.9°, monthly | NOAMADS, Large and Yeager (2009) |

Expansion of Acronyms: WOA = World Ocean Atlas, SeaWIFS = Sea-Viewing Wide Field-of-View Sensor, GEBCO = General Bathymetric Chart of the Oceans, NOAMADS = NOAA National Operational Model Archive and Distribution System. (*) Three available Mixed layer depth (MLD) definitions in WOA (vd=variable potential density, pt=potential temperature, pd=potential density) were processed from csv to NetCDF and extracted. Following Chance et al (2014), the monthly sum and maximum MLD was also computed (vd, pt, pd) and used for building predictions of iodide. When the variable just MLD is shown, it is MLD as defined by potential temperature.

**Table 2.** Splits of dataset used to evaluate outliers and their performance against the withheld data. The Root Mean Square Error (RMSE) statistic given as the mean of the performance against the withheld data for 20 different models built from 20 different pseudo-random initialisations (Sect 3.2). The model used here includes ancillary variables of temperature, depth and salinity which were thought to intuitively give a reasonable result. "#" gives the number of samples in each dataset.

| Description | Mean RMSE vs. model (withheld data), nM | # |
|---|---|---|
| Just coastal & no outliers | 35.8 | 819 |
| Salinity ≥30 PSU & no outliers | 36.7 | 1278 |
| No Skagerrak or outliers | 37.3 | 1293 |
| No outliers | 37.6 | 1306 |
| Just non-coastal & no outliers | 40.4 | 487 |
| All | 95.1 | 1342 |



**Table 3.** Statistics for observations and predictions by the ensemble prediction ("RFR(ensemble)") and existing parameterisations against the entire dataset of observations. The Root Mean Square Error (RMSE) is all shown against the entire and the withheld data.

| | Mean nM | SD nM | 25% nM | median nM | 75% nM | RMSE (withheld), nM | RMSE (entire), nM |
|---|---|---|---|---|---|---|---|
| Obs. | 94.8 | 67.2 | 36.8 | 85 | 140 | - | - |
| RFR(Ensemble) | 95.6 | 60.3 | 41.5 | 89.4 | 139.1 | 34.7 | 23 |
| Chance et al. (2014) | 93.7 | 60 | 38.3 | 86.2 | 149.8 | 50.2 | 49.2 |
| MacDonald et al. (2014) | 39.7 | 30.5 | 13.1 | 32.1 | 66.1 | 75.3 | 75 |

**Table 4.** Statistics on predicted global annual sea-surface values from new and existing parameters at a horizontal resolution of 0.125°x0.125°. Existing parameterisations (Chance et al., 2014; MacDonald et al., 2014) and the ensemble prediction are shown in bold.

| | mean | std. dev. | 25% | median | 75% | max |
|---|---|---|---|---|---|---|
| **Chance et al. (2014)** | 128.1 | 64.9 | 49.1 | 122.1 | 179.4 | 226.2 |
| **MacDonald et al. (2014)** | 58.9 | 34.9 | 17.1 | 50.5 | 86.5 | 125.7 |
| **RFR(Ensemble)** | 105.8 | 45.6 | 51.5 | 106 | 138.5 | 220.4 |
| RFR(TEMP+NO3+MLD+SAL) | 108.8 | 44.2 | 62.1 | 105 | 141.8 | 208.6 |
| RFR(TEMP+SWrad+NO3+MLD+SAL) | 108.5 | 43.1 | 61.5 | 108.5 | 140.7 | 197.3 |
| RFR(TEMP+DEPTH+NO3+SWrad) | 106.9 | 47.6 | 48.1 | 108.5 | 140.9 | 227.3 |
| RFR(TEMP+DEPTH+SAL+SWrad) | 106.6 | 46.9 | 50.1 | 109.5 | 137.1 | 241.3 |
| RFR(TEMP+DEPTH+NO3) | 106.2 | 48.6 | 50.4 | 103.6 | 140.7 | 234.5 |
| RFR(TEMP+DEPTH+SAL) | 105.8 | 47.2 | 52.3 | 103.9 | 134.1 | 248.4 |
| RFR(TEMP+DEPTH+SAL+Phos) | 105.1 | 47.5 | 51.6 | 98.8 | 138.3 | 242.7 |
| RFR(TEMP+DEPTH+SAL+NO3) | 104.9 | 47.2 | 52.7 | 103.9 | 136.2 | 233.5 |
| RFR(TEMP+DEPTH+SAL+ChlrA) | 102.8 | 47.4 | 48 | 98.3 | 135 | 256 |
| RFR(TEMP+DEPTH+ChlrA) | 102.3 | 47.4 | 46.1 | 96.7 | 136.2 | 254 |





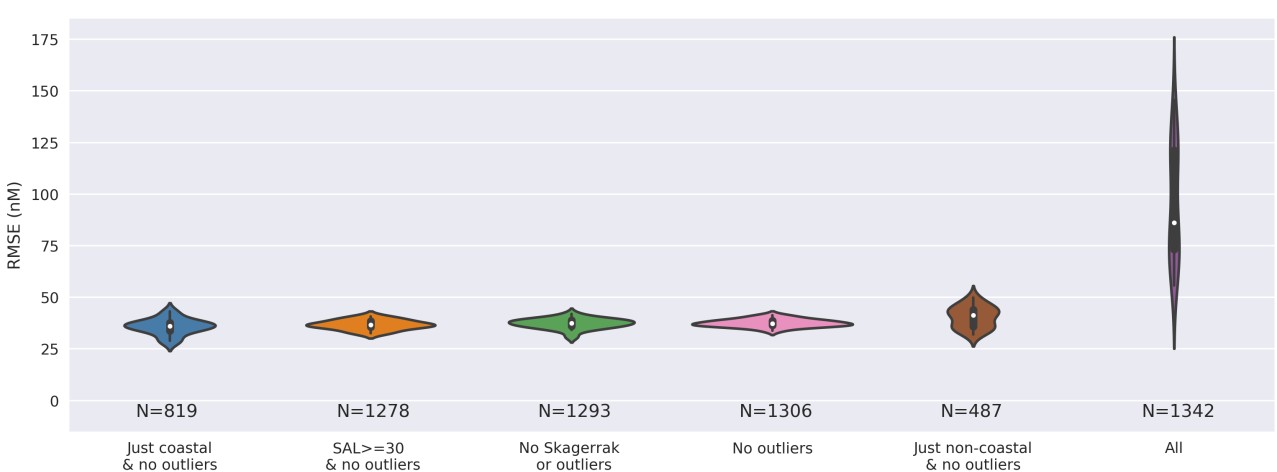

**Figure A1.** Combined kernel density and boxplots ("violin plots") showing the distribution of Root Mean Square Error (RMSE) for 20 different models built from 20 different pseudo-random initialisations for different selection of the dataset as described in Table 2 and Sect 3.2. Models built using the whole dataset ("all"), including outliers, show a significantly higher RMSE due to observations with higher iodide concentrations. The model used here includes ancillary variables of temperature, depth and salinity which were thought to intuitively give a reasonable result.





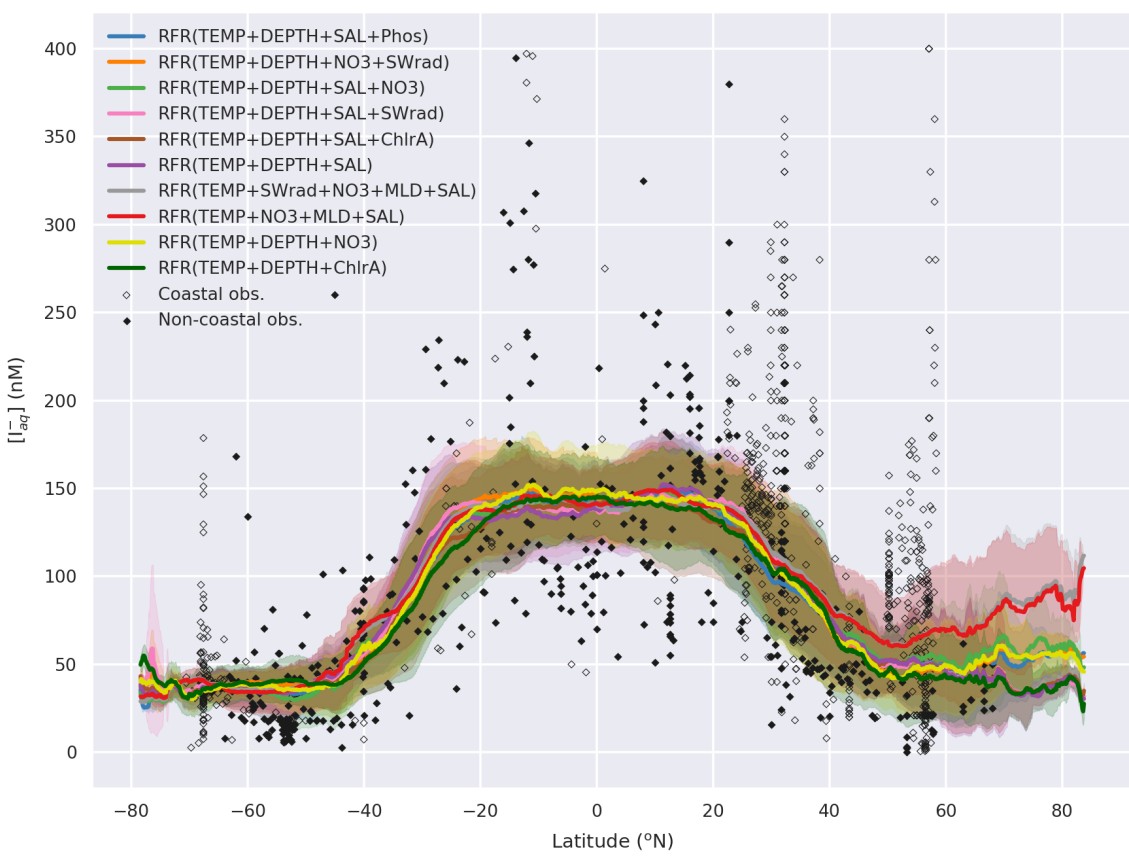

**Figure A2.** Predicted global sea-surface iodide for all ensemble members plotted against latitude, overlaid with observed concentrations. Shaded regions give (±) the average standard deviation for a given latitude. The standard deviation is the monthly standard deviation for a single ensemble members ("RFR(Ensemble)") or within a single prediction for existing parameterisations (Chance et al., 2014; MacDonald et al., 2014). Filled diamonds show non-coastal observations and unfilled ones show coastal values. Extent of X axis is shown for grid-boxes that are entirely water.

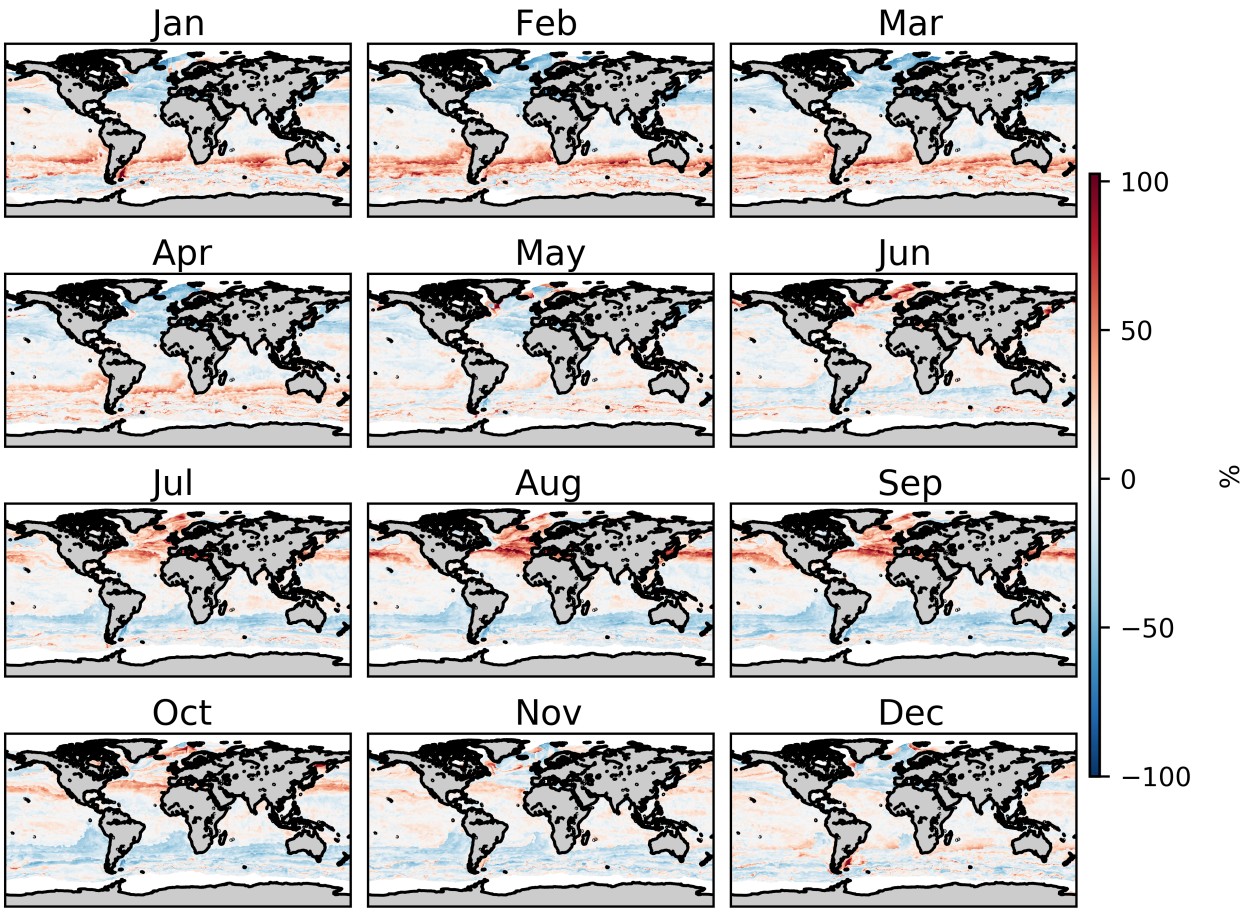

**Figure A3.** Percentage difference in monthly sea-surface iodide from the annual mean field predicted by the ensemble. Only locations that are entirely water are included.





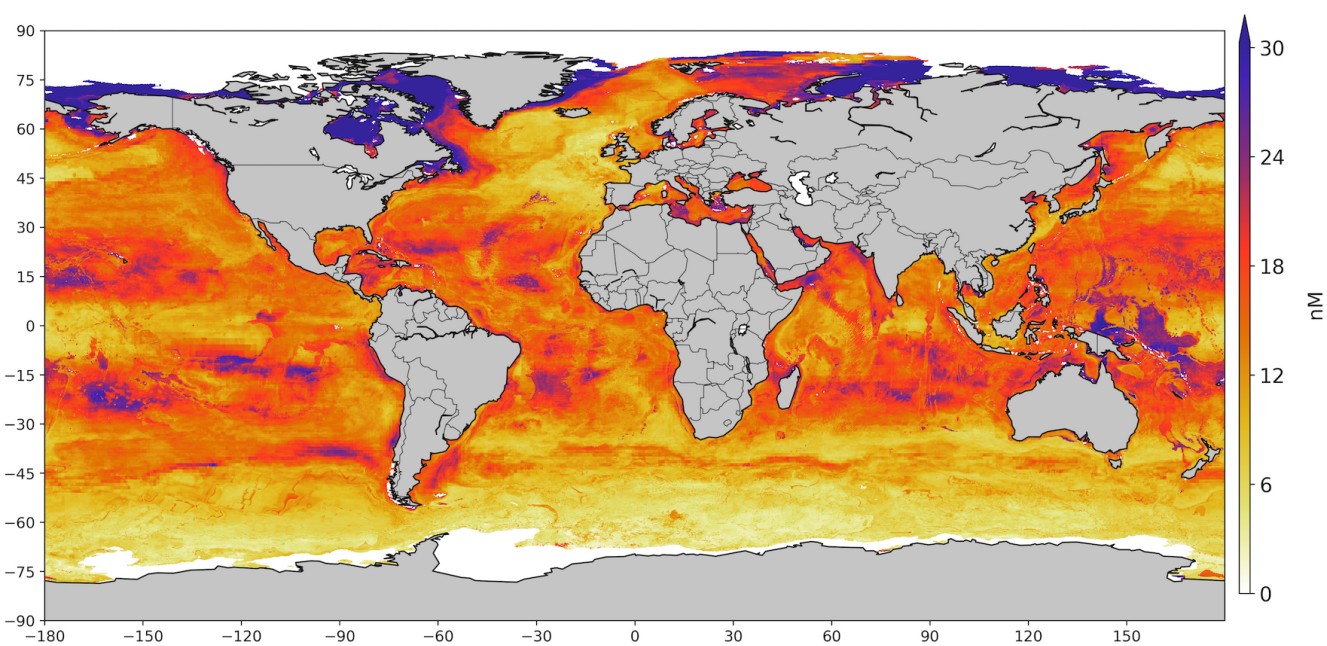

**Figure A4.** Annual average spatial uncertainty in predicted sea-surface iodide for the ensemble of models. Spatial variation was calculated as the standard deviation in monthly average values for all models. Values are limited to 30nM for contrast and the maximum value plotted is 52 nM in the Northern high-latitudes. Only locations that are entirely water are included in spatial the average.

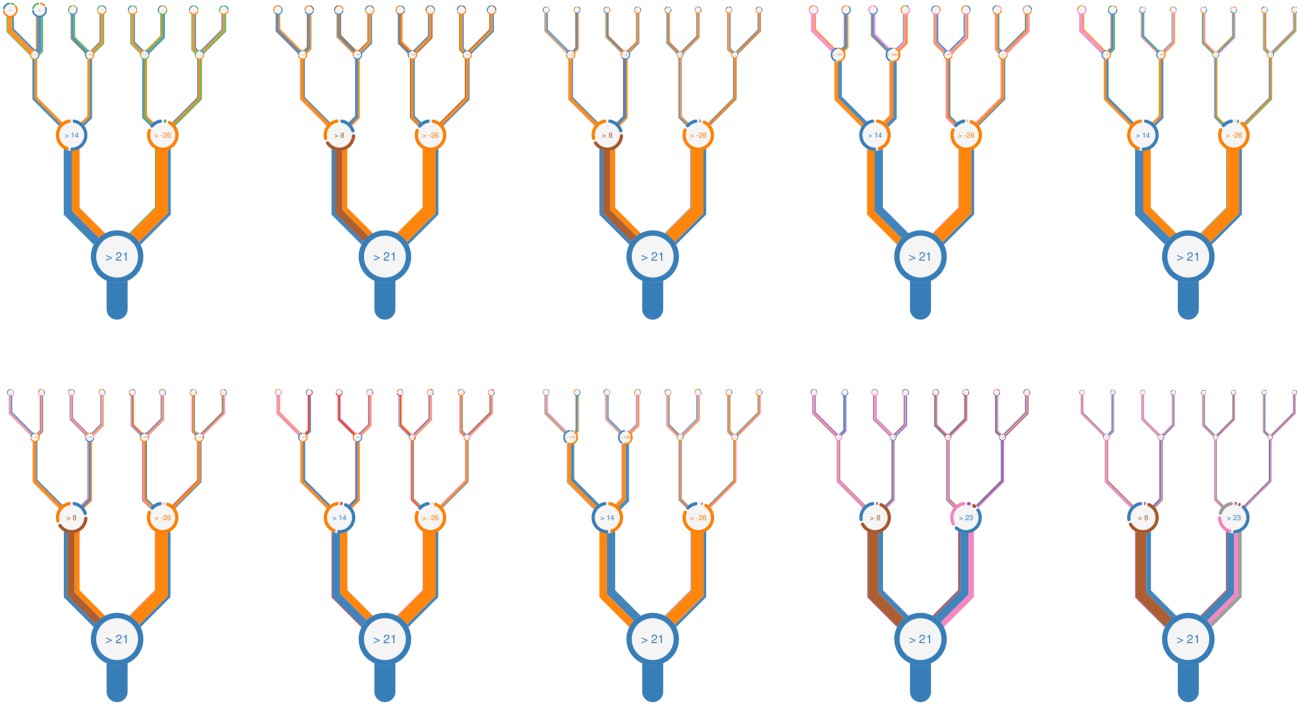

**Figure A5.** Representation of all forests within the ensemble (also shown as thumbnails in Fig. 3c). The branches are coloured by percentage of each variable that drives the decision at a given node. Thickness of branches gives the average throughput of the dataset through a given node. Variable names are coloured as per the following coloured text: temperature (blue, °C), depth (orange,metres), chlorophyll-A (green, mg m$^{-3}$), salinity (pink, PSU), nitrate (brown, $\mu$g m$^{-3}$), mixed layer depth (MLD; purple, metres), phosphate (red, $\mu$g m$^{-3}$), and shortwave radiation (grey, Wm$^{-2}$)



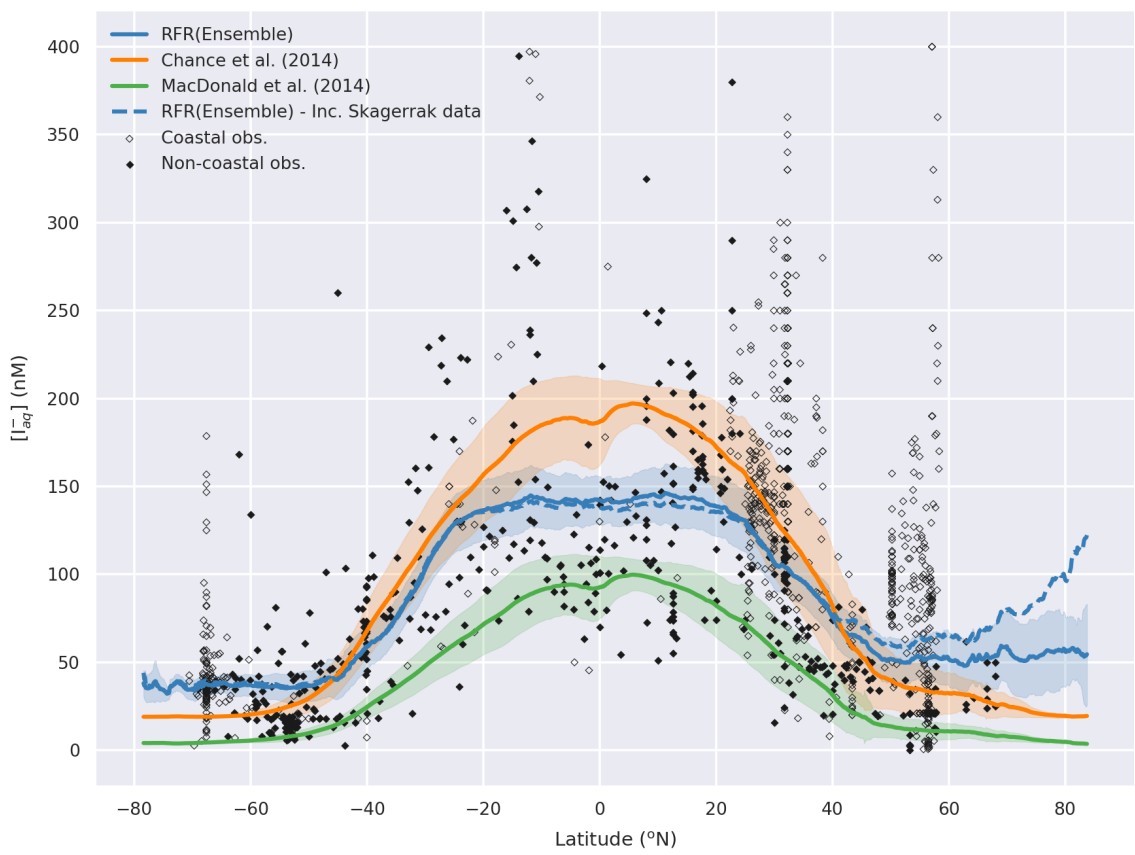

**Figure A6.** Predicted latitudinal average sea-surface iodide plotted against latitude, overlaid with observed concentrations. Figure is equivalent to Fig 7, but the dashed line shows the prediction including data from the Skagerrak strait (Truesdale et al., 2003) . Solid lines give mean values and shaded regions give $\pm$ the average standard deviation. For the ensemble the standard deviation is the monthly standard deviation within all ensemble members. Filled diamonds show non-coastal observations and unfilled ones show coastal values. Extent of x axis is shown for grid-boxes that are entirely water.



**Table A1.** Summary of sea-surface iodide parameterisations in global and regional atmospheric models

| Model | Parameterisation | Refs |
|---|---|---|
| CMAQ | Observed range | Chang et al. (2004) |
| CMAQ | Linearly fitting Chlorophyll-a to $[I_{aq}^-] <$ range of 100-400 nM | Oh et al. (2008) |
| REMOTE | Observed range | Coleman et al. (2010) |
| MESSy-ECHAM5 | Campos $NO_3^-$ relationships | Ganzeveld et al. (2009) |
| CAM-Chem | Eqn 2 | Prados-Roman et al. (2015); Saiz-Lopez et al. (2014) |
| CMAQ | Eqn 2 | Gantt et al. (2017); Sarwar et al. (2016, 2015) |
| GEOS-Chem | Eqn 1 | Sherwen et al. (2016a) |
| GEOS-Chem | Eqn 2 | Sherwen et al. (2016c, b, 2017a, b) |
| ACCESS-UKCA | Eqn 2 | Luhar et al. (2017, 2018) |



**Table A2.** Statistics on observations and predicted values from the new ensemble and existing parameterisations at locations of observations. Root Mean Square Error (RMSE) is shown against the withheld data and the entire dataset of observations. Ensemble members, ensemble prediction ("RFR(Ensemble)"), and existing parameterisations shown in bold. Values are shown for all 38 models built, including those not included in the ensemble.

| | mean | std. dev. | 25% | median | 75% | RMSE (withheld) | RMSE (entire) |
|---|---|---|---|---|---|---|---|
| **Obs.** | 94.8 | 67.2 | 36.8 | 85 | 140 | - | - |
| **MacDonald et al. (2014)** | 39.7 | 30.5 | 13.1 | 32.1 | 66.1 | 75.3 | 75 |
| **Chance et al. (2014)** | 93.7 | 60 | 38.3 | 86.2 | 149.8 | 50.2 | 49.2 |
| **RFR(Ensemble)** | 95.6 | 60.3 | 41.5 | 89.4 | 139.1 | 34.7 | 23 |
| **RFR(TEMP+DEPTH+SAL+Phos)** | 95.4 | 60.8 | 41.3 | 89.2 | 139.9 | 33.2 | 22.6 |
| **RFR(TEMP+DEPTH+NO3+SWrad)** | 95.5 | 60.4 | 40.8 | 89.2 | 139.8 | 34.9 | 23.3 |
| **RFR(TEMP+DEPTH+SAL+NO3)** | 95.5 | 60.6 | 41.8 | 89 | 140.2 | 35.2 | 23.3 |
| **RFR(TEMP+DEPTH+SAL+SWrad)** | 95.3 | 60.6 | 41.7 | 89.5 | 138.7 | 35.3 | 23.4 |
| **RFR(TEMP+DEPTH+SAL+ChlrA)** | 95.7 | 60.5 | 41.7 | 89.4 | 138.1 | 35.6 | 23.4 |
| **RFR(TEMP+DEPTH+SAL)** | 95.5 | 60.6 | 41.3 | 89.4 | 137.9 | 35.9 | 23.7 |
| **RFR(TEMP+SWrad+NO3+MLD+SAL)** | 95.6 | 60.7 | 41.1 | 89.4 | 140.9 | 36.2 | 24.1 |
| **RFR(TEMP+NO3+MLD+SAL)** | 95.8 | 60.6 | 41.3 | 88.9 | 140.9 | 36.6 | 24.3 |
| **RFR(TEMP+DEPTH+NO3)** | 95.6 | 60.5 | 41 | 89.3 | 140.4 | 36.9 | 24 |
| **RFR(TEMP+DEPTH+ChlrA)** | 95.8 | 60.3 | 41.3 | 89.4 | 139.9 | 37.2 | 24.3 |
| RFR(TEMP+DEPTH) | 95.6 | 60.6 | 41.1 | 88.6 | 139.2 | 37.4 | 24.4 |
| RFR(TEMP+SAL+NO3) | 95.6 | 60.6 | 42 | 89.3 | 141 | 37.5 | 24.6 |
| RFR(SWrad+SAL+DEPTH) | 95.5 | 58.2 | 46.3 | 89.4 | 135.5 | 37.5 | 24.7 |
| RFR(SWrad+SAL+NO3) | 95.7 | 59.8 | 42.7 | 90.4 | 137.4 | 38.3 | 25.2 |
| RFR(NO3+SWrad) | 95.6 | 58.8 | 41.8 | 94.6 | 136.9 | 38.9 | 27.9 |
| RFR(TEMP+SAL) | 95.5 | 60.5 | 41.4 | 88.7 | 140.2 | 39.8 | 25.6 |
| RFR(TEMP+NO3) | 96 | 60.5 | 41.5 | 88.1 | 139.9 | 40.3 | 25.9 |
| RFR(DEPTH+SAL) | 95.6 | 57.6 | 47.7 | 90.2 | 134.6 | 40.9 | 26.7 |
| RFR(NO3+SAL) | 95.6 | 58.9 | 42.7 | 90.3 | 138.9 | 43 | 27.5 |
| RFR(Phos) | 95.4 | 57.6 | 44.1 | 92.2 | 141.1 | 43.6 | 31.3 |



**Table A3.** Table A2 continued.

| | mean | std. dev. | 25% | median | 75% | RMSE (withheld) | RMSE (all) |
|---|---|---|---|---|---|---|---|
| RFR(O2) | 95.7 | 60.5 | 40.2 | 89.3 | 141.9 | 43.7 | 30.1 |
| RFR(SWrad+SAL) | 95.2 | 59 | 46.2 | 89.4 | 136.6 | 43.8 | 27.3 |
| RFR(TEMP) | 95.7 | 60.1 | 42.3 | 89.2 | 139.3 | 45.2 | 28.6 |
| RFR(NO3) | 95.6 | 58.6 | 43.5 | 90.2 | 141.1 | 46.6 | 32.1 |
| RFR(MLDpd_max) | 95.2 | 56.1 | 47 | 100.3 | 138.3 | 47.1 | 36.2 |
| RFR(MLDpt_max) | 95.1 | 54.4 | 45.6 | 100.3 | 138.3 | 47.2 | 38.5 |
| RFR(SWrad) | 95.7 | 58.7 | 47.2 | 89.4 | 137.5 | 47.5 | 32 |
| RFR(MLDpd_sum) | 95.3 | 57.2 | 47.6 | 93.9 | 141.2 | 48.3 | 34.3 |
| RFR(MLDpt_sum) | 95.7 | 57.6 | 46.5 | 94.3 | 140.2 | 49.1 | 34.5 |
| RFR(MLDvd_sum) | 95.2 | 55.8 | 47.9 | 91.8 | 139.6 | 49.1 | 35.4 |
| RFR(Sil) | 95.4 | 57.6 | 45.2 | 90.4 | 140 | 49.5 | 33 |
| RFR(MLDvd_max) | 94.9 | 52.5 | 47.9 | 99.1 | 134.1 | 51.5 | 40.8 |
| RFR(MLDpt) | 95.4 | 53.9 | 49.7 | 94.3 | 134 | 54.9 | 40.9 |
| RFR(MLDpd) | 95.8 | 52.2 | 55.5 | 91.7 | 126.4 | 55.2 | 41.8 |
| RFR(SAL) | 95.9 | 56.1 | 52.1 | 89.3 | 132.7 | 56.8 | 34.6 |
| RFR(DEPTH) | 96.5 | 53 | 51.7 | 90.8 | 135.3 | 57 | 39.6 |
| RFR(MLDvd) | 95.8 | 49.2 | 53.9 | 87.4 | 134.1 | 57.4 | 45.2 |
| RFR(ChlrA) | 95.6 | 53.4 | 53.9 | 89.6 | 131.6 | 59.6 | 36.1 |

**Table A4.** Statistics on observations and predicted values by ensemble and existing parameterisations at locations of observations, but just for the withheld dataset locations. Table 3 shows the values for both withheld and entire dataset.

| | mean | std. dev. | 25% | median | 75% | RMSE |
|---|---|---|---|---|---|---|
| Obs. | 94.3 | 67.4 | 36.9 | 84 | 138.6 | - |
| RFR(Ensemble) | 96.7 | 57.8 | 47.2 | 93.1 | 137.4 | 34.7 |
| Chance et al. (2014) (Eqn. 1) | 93.6 | 59.7 | 39 | 87.9 | 148.4 | 50.2 |
| MacDonald et al. (2014) (Eqn. 2) | 39.6 | 30.2 | 13.4 | 32.8 | 65.1 | 75.3 |



**Table A5.** Spatial and temporal resolutions of global monthly iodide fields available for download from the United Kingdom's Centre from Environmental Data Analysis (Sherwen et al. (2019); DOI:https://doi.org/10/gfv5v3). Regridding was performed in Python using the open-source xESMF package (Zhuang, 2018).

| Resolution | | Bottom Left grid edge | |
| Lat. x Lon. | Model/description | Lon. | Lat. |
| --- | --- | --- | --- |
| 0.125°x0.125° | GEOSChem in GEOS5 (Hu et al., 2018) | -180.0625 | -90.0625 |
| 1°x1° | Centred on unit degrees | -180.5 | -90.5 |
| 1°x1° | Centred on 0.5 | -180 | -90 |
| 2°x2.5° | GISS ModelE (Miller et al., 2014) | -178.75 | -90 |
| 2°x2° | ACCMIP (Lamarque et al., 2013) | -180 | -90 |
| 2°x2.5° | GEOSChem (Bey et al., 2001) | -181.25 | -91 |
| 2.5°x3.75° | UKCA (O'Connor et al., 2014) | -180 | -90 |
| 4°x5° | GEOSChem (Bey et al., 2001) | -182.5 | -92 |

Latitudes less than -90° indicate half-boxes at poles. Acronyms expand to: United Kingdom Chemistry and Aerosols (UKCA), Atmospheric Chemistry and Climate Model Intercomparison Project (ACCMIP), and Goddard Institute for Space Studies (GISS).