# Peer review of "A machine learning based global sea-surface iodide distribution"

_Earth System Science Data, 2019_

## Referee Comment (RC1) · Peer Johannes Nowack (Referee) · 23 Apr 2019

**General comments:**

The paper by Sherwen et al. introduces a new dataset for monthly-mean sea-surface iodide concentrations. Their new machine learning approach to create the dataset is both appealing and promising because it can simultaneously account for observationally-constrained relationships between several predictors and iodide in an objective manner while capturing potentially complex functional dependencies. I find their approach interesting not just for the creation of this new dataset (which indeed could be used widely in atmospheric chemistry studies), but also for inference. For example, their work could motivate further research into the physical and biological

drivers of iodide changes at the sea-surface, or measurement campaigns in certain world regions. As such, the study could be of much broader interest than just for the creation of a new dataset.

Overall, the paper is well-written, easy to follow and scientifically thorough. It deserves rapid publication subject to some mostly very minor revisions/suggestions listed below.

The datasets discussed in the paper are indeed accessible through the provided link in the standard netCDF format.

**Specific comments:**

- In the abstract and main text: for readers less accustomed to global iodide datasets it would be good to explain in somewhat more detail the recommended application context of this dataset. Could it be used to represent iodine emissions in historical, or even future, climate change simulations (where e.g. SSTs are subject to change and have in fact already changed) or ozone hole studies, or is it really applicable only to present-day air quality studies? What are the general assumptions here given that you create a non-transient monthly-mean climatological dataset? Are there any transient effects in the training datasets and what time period have the observations been sampled over? Your citation implies a period from 1967-2018, but it would be good to state this explicitly.

- Abstract: I would say specifically that the sample size has increased by 45% to avoid misunderstandings.

- p.3/4, beginning of section 2: here might be a good place to state the time period and to mention that you make the approximation that the relationships are stationary (?).

- Section 2: in the abstract you mention the use of climatological ancillary fields. You don't specify a time period for the observations either: 'For each iodide observation, the nearest point in space and time was extracted from the high resolution gridded ancillary data. For the 31 iodide observations where a month was not available (Luther and Cole, 1988; Tsunogai Shizuo and Henmi, 1971; Wong and Cheng, 1998), an arbitrary month was chosen (of March for Northern hemispheric observations and September for Southern hemispheric observations)'. Does this mean that you simply regress iodine observations against the SST etc fields purely based on the seasonal climatology? Why would you not use the temperature at the actual time when the iodide sample was taken? In addition, why would one not just archive the random forest regressor and use this model to predict SST etc consistent iodide concentration interactively in simulations (consistent with the actual state simulated by the model)? Could you discuss these aspects briefly; not necessarily in the main paper but maybe in reply to this comment?

- p. 4 l.26: similar - just for clarification in this review; what is meant by a month was not available? The observations exist, but no corresponding time reference?

- p.4 l.5: it is a question of taste, but I would somehow prefer predictors, regressors, input variables, input features etc over independent variables, which can sometimes be misunderstood (even though not incorrect), see e.g. discussion here: https://stats.stackexchange.com/questions/357745/in-regression-analysis-why-do-we-call-independent-variables-independent [...] I definitely don't feel strongly about this, so this is entirely up to the authors to decide (ie they can leave as is).

- p.5 l.26: a reference for the stratified sampling approach or more detailed description possible?

- p.7: it is not entirely clear to me at this point in the paper if the RMSE improvement after outlier removal is due to (a) the outliers being removed prior to training (are not involved at all), or (b) due to the outliers being removed from the validation/test

data so that the error on these specific predictions is simply not included in the final evaluation (i.e. the algorithm is simply not good at predicting those large value outliers). I guess the last sentence of this section implies (a) is the case here, but maybe good to say explicitly in the same sentence (I later also noticed that you discuss the alternatives below, but better to clarify this aspect here, too).

- p.8 l.3: so this becomes effectively an ensemble of an ensemble method (which random forests are)? Not sure if some people could misunderstand that given that you mention random forests as an ensemble method in Figure caption 3; you might consider using another term than ensemble here? You can leave as is though.

- p.9 section 4.2: Could the features associated with deep bathymetry (see your Discussion on p.11 l20) be down to a non-realistic assumption of the importance of bathymetry in those regions (based on a biased training dataset)? A simple test would be to check the predictions of the best performing models that do not include DEPTH; do those also predict such structure? If not, it could imply that those models actually show better physical generalizability (as far as we know) and could, therefore, be the preferable option. There might simply not be enough measurements in the training set covering grid coordinates along the Atlantic Ridge and as a result, it does not show up as an important error contribution in the training dataset.

- Did you retrain your forest on the entire available observational dataset before making the final predictions using the best performing models during the cross-validation procedure? This might be advantageous because you would take into account all available observations in training your algorithm (while not changing any other tunable parameters).

- p. 10 l.35: Relative (?) uncertainties are largest. . .

- p.12 l.9-13: this could be misunderstood. Do you mean by 'trend' a spatial pattern? It kind of links to my question about the consideration of transient effects and both aspects could be discussed here.

- In our uploaded .nc files, there appears to be no mask over land surfaces even though you only provide iodide data for the sea-surface? Can you explain? How are these values to be interpreted by modellers?

- A1, p. 13 l.13: This could be an interesting feature to explore with other regression models which allow for extrapolation outside the training domain. I guess this 'flat' prediction could be due to the fact that the random forest hasn't seen many inputs representative of this area yet (e.g. in terms of SSTs)? Maybe looking at how predictor-output relationships behave at the boundaries (can it be extrapolated) would be promising? Not necessarily something to be considered for this paper, but for future data updates (i.e. just a thought that may be ignored).

- Figure 7: for consistency, wouldn't it make more sense to plot the average plus standard deviation of the observations as well? Currently, the comparison seems rather unfair towards the parameterisations and emphasizes high values in the observations that deviate much from the predictions.

**Technical corrections/typing errors:**

- p.2, formula (2): I know this is a unit conversion, but the extra 10e9 multiplication reads like a mistake. Would summarise the two factors into a single multiplication factor.

- p.3 l4: non-coastal

- p.3 l8-10: The choice of parameterisation (Eqn. 2 versus Eqn. 1) results in a difference of 50

- p.3 l16: formulation

- p.4 l23: typo; this is not described in section 2, but in section 3.3.

- p.5 l29: typo

- p.5 l35: revise sentence "All forests..."

- p.6 l10-12: sentence is difficult to read.

- p.6 l29: typo

- p.7 l18: typo

- p.10 l16: typo

- p.10 l32: typo

- p.12 l.10: typo

- Figure 6 caption: revise last sentence

---

## Referee Comment (RC2) · Laurens Ganzeveld (Referee) · 26 Jun 2019

The paper describes compilation of a global ocean water Iodide climatology applying a machine learning approach that combines a compilation of Iodide observations and other climatologies on parameters such as SST, nitrate, radiation to explain these observations. This compilation of a global ocean water Iodide climatology is of large relevance for large-scale studies on air quality, atmospheric chemistry and climate interactions given the role of ocean Iodide in emissions to the atmosphere affecting atmospheric composition and also involving some potentially relevant feedback mechanisms. Overall the paper is well written and presents a sound approach to provide a new dataset to be further applied in Earth system studies and fits within the scope of ESSD. Consequently, I recommend publication of this manuscript in ESSD after the

following generally minor comments have been addressed. Note that, since I am not experienced with machine learning methods, my comments are mostly limited to the context of the presented work and the description of the main results coming out of this approach.

Abstract: "simple functions of sea-surface temperature (Chance..)"; I would leave out here the references (generally not included in abstract) and rather state that: "have generally fitted sea-surface iodide observations to relatively simple functions using iodide proxies such as nitrate and sea-surface temperature"

Page 2: line 4, "..oxygen level."

Page 2: line 10: I am generally not keen on calling for inclusion of references to my work but since this reference is already included in this ms, the study by Ganzeveld et al. (2009) was also mainly aiming to assess the role of Iodide as one of main reactants in oceanic O3 deposition and the resulting impacts on atmospheric composition.

Page 2, line 16: "catalytically destroy ozone (Chameides and Davis, 1980)." Here it would be interesting to add here that this thus mechanisms thus implies the presence of a negative feedback mechanism involving this O3 and Iodide chemistry as also being assessed in a modelling study by Prados Roman et al. (2016)

Page 4, line 7/8: "they need to be available at an appropriate resolution as a gridded product"; here it would be useful to indicate an estimate of this required resolution given the (known) scale of the heterogeneity in the distribution of the parameters that potentially explain Iodide. For example, given the anticipated (large) contrasts between coastal and open ocean waters, what minimum resolution is needed?

Page 4; line 14: " This horizontal resolution was used as this is the highest resolution of the current generation of global atmospheric chemistry simulations (Hu et al., 2018)." Here it might be interesting to mention that this resolution of 12.5km also seems to be sufficient for application meso-scale meteorological – Air quality model studies used

for regional scale studies. We deploy for example now the meso-scale modelling system WRF-CHEM at a resolution of ∼20km, including a mechanistic representation of oceanic ozone deposition including Iodide reactivity.

Section 3.1: Not being very familiar with the application of machine learning methods, I really appreciate the explanation that this given on the specifics of the approach. There is still some terminology that would require further in-depth checking out the details of the followed approach but think that is a nice way to also explain it all to the readers mostly interested at the end in the final outcome of application of this methodology, the global Iodide dataset. Page 9, line 35: "…variability is both well constrained by observations. Some of the highest…"

Page 10, lines 12-14: " The new predicted values lay between Chance et al. (2014) and MacDonald et al. (2014) in the tropics, however, within the polar regions, the new prediction is significantly higher than both of the previous parameterisations." Not so much a comment but so this result is further stressing the need for additional measurements in the Arctic that we might now get with the upcoming MOSAiC field campaign.

Page 11, line 26/27: "A higher iodide sea-surface concentration would also result in a greater calculated ozone deposition (Luhar et al., 2017; Sarwar et al., 2016).". Here a reference to the Ganzeveld et al. 2009 paper would be really appropriate with this paper showing the first step to consider the impact of global Iodide distribution on global ozone deposition (and atmospheric ozone).

Page 11, line 31: "Considering that the average predicted concentration globally here is 106 nM (Sect. 4.2), these errors are notable"

Figure 6: here the observations are indicated by dots that are so small that you cannot see to what extent the inferred values compare to those observations. You could try to enhance the size of those dots.

---

## Author Comment (AC1) · 24 Jul 2019

We thank both reviewers for the positive and constructive comments on our manuscript. We have updated the manuscript accordingly and responded directly to all comments raised below. We are grateful for their time and input which we feel has improved our manuscript.

Additionally for the sake of transparency and reproducibility, the code used for this work has been separately archived with a DOI and text has been added to link to this in the methods section. The links to outputted data archived with CEDA have also been updated to the latest archived version (v0.0.1), which includes files regridded to further resolutions (e.g. for use in the CMAQ air quality model). Some very minor text changes

were also made for clarity as well as a few minor typographic error updates that do not affect the conclusions or discussion.

Reviewer #1 - Peer Johannes Nowack (Referee) General comments The paper by Sherwen et al. introduces a new dataset for monthly-mean seasurface iodide concentrations. Their new machine learning approach to create the dataset is both appealing and promising because it can simultaneously account for observationally-constrained relationships between several predictors and iodide in an objective manner while capturing potentially complex functional dependencies. I find their approach interesting not just for the creation of this new dataset (which indeed could be used widely in atmospheric chemistry studies), but also for inference. For example, their work could motivate further research into the physical and biological drivers of iodide changes at the sea-surface, or measurement campaigns in certain world regions. As such, the study could be of much broader interest than just for the creation of a new dataset.

Overall, the paper is well-written, easy to follow and scientifically thorough. It deserves rapid publication subject to some mostly very minor revisions/suggestions listed below.

The datasets discussed in the paper are indeed accessible through the provided link in the standard netCDF format.

We are grateful for reviewer #1's positive response to our manuscript, comments on broader transferability of the approach and value to broader community, and recommendation for rapid publication.

We respond directly to the specific comments below:

Specific comments • In the abstract and main text: for readers less accustomed to global iodide datasets it would be good to explain in somewhat more detail the recommended application context of this dataset. Could it be used to represent iodine emissions in historical, or even future, climate change simulations (where e.g. SSTs are subject to change and have in fact already changed) or ozone hole studies, or

is it really applicable only to present-day air quality studies? What are the general assumptions here given that you create a non-transient monthly-mean climatological dataset? Are there any transient effects in the training datasets and what time period have the observations been sampled over? Your citation implies a period from 1967-2018, but it would be good to state this explicitly.

Clarification on the opportunities for using the output and discussion of the non-transient nature of the shared product have been to the main text and abstract.

• Abstract: I would say specifically that the sample size has increased by 45% to avoid misunderstandings.

Updated.

• p.3/4, beginning of section 2: here might be a good place to state the time period and to mention that you make the approximation that the relationships are stationary (?).

Updated.

• Section 2: in the abstract you mention the use of climatological ancillary fields. You don't specify a time period for the observations either: 'For each iodide observation, the nearest point in space and time was extracted from the high resolution gridded ancillary data. For the 31 iodide observations where a month was not available (Luther and Cole, 1988; Tsunogai Shizuo and Henmi, 1971; Wong and Cheng, 1998), an arbitrary month was chosen (of March for Northern hemispheric observations and September for Southern hemispheric observations)'. Does this mean that you simply regress iodine observations against the SST etc fields purely based on the seasonal climatology? Why would you not use the temperature at the actual time when the iodide sample was taken? In addition, why would one not just archive the random forest regressor and use this model to predict SST etc consistent iodide concentration interactively in simulations (consistent with the actual state simulated by the model)? Could you

discuss these aspects briefly; not necessarily in the main paper but maybe in reply to this comment?

In response to other comments from both reviewers, multiple updates have been made to the manuscript which relate to the above comment and somewhat address the points raised (such as including the period of observations at multiple points during the text). Additionally, we reply directly below as requested.

Monthly climatologies were used for all ancillary variables. The reasons for taking this approach include internal consistency (as not all ancillary fields were available at higher temporal resolution, especially when also maintaining spatial resolution - e.g. nitrate), completeness at higher resolution (non-seasonally averaged variables would often require more interpolation for missing values) and the fact that observations (1967-2018) were considered without inter annual variability on a monthly basis too.

In theory, the approach could be used interactively for the ancillary variables (where data is available for a given variable - e.g. through a satellite product for sea-surface temperature). However, germane to the reasons given above, risks of error/considering missing data within these fields could negatively affect predictions. When combined with the lack of knowledge of any temporal changes in iodide (not considered in this work), the increased risks out-weigh the modest gains expected here.

 c p. 4 l.26: similar - just for clarification in this review; what is meant by a month was not available? The observations exist, but no corresponding time reference?

Correct. No month is available within the original literature. Please see the accompanying data descriptor paper on the new observational dataset for more details.

"Global sea-surface iodide observations, 1967-2018",Chance R.; Tinel L.; Sherwen T.; Baker A.; Bell T.; Brindle J.; Campos M.L.A.M.; Croot P.; Ducklow H.; He P.; Hoogakker B.; Hopkins F.E.; Hughes C.; Jickells T.; Loades D.; Reyes Macaya D.A.; Mahajan A.S.; Malin G.; Phillips D.P.; Sinha A.K.; Sarkar A.; Roberts I.J.; Roy R.; Song X.;

Winklebauer H.A.; Wuttig K.; Yang M.; Zhou P.; Carpenter L.J., in review., 2019

• p.4 l.5: it is a question of taste, but I would somehow prefer predictors, regressors, input variables, input features etc over independent variables, which can sometimes be misunderstood (even though not incorrect), see e.g. discussion here: https://stats.stackexchange.com/questions/357745/in-regressionanalysis-why-do-we-call-independent-variables-independent [...] I definitely don't feel strongly about this, so this is entirely up to the authors to decide (ie they can leave as is).

We thank the reviewer for the suggestion but have retained the original terminology presented for ease of comprehension considering the manuscript's target audience is one where machine learning nomenclature may not be common knowledge.

• p.5 l.26: a reference for the stratified sampling approach or more detailed description possible?

The sentences discussing this have been updated.

• p.7: it is not entirely clear to me at this point in the paper if the RMSE improvement after outlier removal is due to (a) the outliers being removed prior to training (are not involved at all), or (b) due to the outliers being removed from the validation/test data so that the error on these specific predictions is simply not included in the final evaluation (i.e. the algorithm is simply not good at predicting those large value outliers). I guess the last sentence of this section implies (a) is the case here, but maybe good to say explicitly in the same sentence (I later also noticed that you discuss the alternatives below, but better to clarify this aspect here, too).

The reviewer is correct in interpreting that approach (a) was taken. This has been clarified in the text.

• p.8 l.3: so this becomes effectively an ensemble of an ensemble method (which random forests are)? Not sure if some people could misunderstand that given that you mention random forests as an ensemble method in Figure caption 3; you might

consider using another term than ensemble here? You can leave as is though.

This is correct. The term ensemble has been retained but it has been clarified in the text that this means the prediction is a prediction from ten-member ensemble which themselves are ensemble predictions.

• p.9 section 4.2: Could the features associated with deep bathymetry (see your Discussion on p.11 l20) be down to a non-realistic assumption of the importance of bathymetry in those regions (based on a biased training dataset)? A simple test would be to check the predictions of the best performing models that do not include DEPTH; do those also predict such structure? If not, it could imply that those models actually show better physical generalizability (as far as we know) and could, therefore, be the preferable option. There might simply not be enough measurements in the training set covering grid coordinates along the Atlantic Ridge and as a result, it does not show up as an important error contribution in the training dataset.

We agree with the reviewer that this may be an artifact caused by dataset sampling and that there may just not be enough observations in these regions (e.g. on along the Atlantic Ridge). Conceptually, depth has been included in the provided parameterisations to be combined with other variables to infer the "coastal" nature of a location. Although choosing the top ten models which do not include depth as an input variable removes the minor imprinting of deep-ocean bathymetric features, it leads to a decrease in skill in predicting the withheld data - increasing it the Root Mean Square Error (RMSE) by 5.3 %. The largest decrease in prediction ability is seen against noncoastal sites, where the RMSE increases by 6.9 %. We have therefore retained depth as variable used in this prediction. We are keen to reapproach this prediction and the variables used once more data is available which would hopefully reduce the effect of biases in the dataset. We hope that publishing this paper may be able to stimulate the community to collect more data.

• Did you retrain your forest on the entire available observational dataset before

making the final predictions using the best performing models during the crossvalidation procedure? This might be advantageous because you would take into account all available observations in training your algorithm (while not changing any other tunable parameters).

The exposure of the models to the testing dataset was minimized as a priority. This was at least in part due to the small size of the dataset used here (∼1300), so a cross-validation approach on the selected top ten models was not performed. As the observational dataset grows, techniques like cross-validation will be again considered.

• p. 10 l.35: Relative (?) uncertainties are largest. . .

Updated.

• p.12 l.9-13: this could be misunderstood. Do you mean by 'trend' a spatial pattern? It kind of links to my question about the consideration of transient effects and both aspects could be discussed here.

Updated.

• In our uploaded .nc files, there appears to be no mask over land surfaces even though you only provide iodide data for the sea-surface? Can you explain? How are these values to be interpreted by modellers?

Further information has been added in the data availability section.

• A1, p. 13 l.13: This could be an interesting feature to explore with other regression models which allow for extrapolation outside the training domain. I guess this 'flat' prediction could be due to the fact that the random forest hasn't seen many inputs representative of this area yet (e.g. in terms of SSTs)? Maybe looking at how predictor-output relationships behave at the boundaries (can it be extrapolated) would be promising? Not necessarily something to be considered for this paper, but for future data updates (i.e. just a thought that may be ignored).

We thank the reviewer for sharing his thoughts here and will consider this when updating the dataset when once more observations are available.

• Figure 7: for consistency, wouldn't it make more sense to plot the average plus standard deviation of the observations as well? Currently, the comparison seems rather unfair towards the parameterisations and emphasizes high values in the observations that deviate much from the predictions.

This would show an average and deviation biased towards the regions sampled by the observations, which would not be a fair comparison to the global sea-surface averages provided for the parameterisations. The plot already contains a lot of information and further lines would increase the difficulty for the reader. For these reasons, this additional line was not be added.

Technical corrections/typing errors: • p.2, formula (2): I know this is a unit conversion, but the extra 10e9 multiplication reads like a mistake. Would summarise the two factors into a single multiplication factor.

Formatting retained for comparability with original manuscript [Macdonald et al. 2014].

• p.3 l4: non-coastal

Hyphen added.

• p.3 l8-10: The choice of parameterisation (Eqn. 2 versus Eqn. 1) results in a difference of 50

Word "different" removed.

• p.3 l16: formulation

Updated.

• p.4 l23: typo; this is not described in section 2, but in section 3.3.

Updated.

• p.5 l29: typo

Revised.

• p.5 l35: revise sentence "All forests..."

Revised.

• p.6 l10-12: sentence is difficult to read.

Revised.

• p.6 l29: typo

Updated.

• p.7 l18: typo

Updated.

• p.10 l16: typo

Revised.

• p.10 l32: typo

Revised.

• p.12 l.10: typo

Revised.

• Figure 6 caption: revise last sentence

Revised.

Reviewer #2 - Laurens Ganzeveld (Referee)

General comments The paper describes compilation of a global ocean water Iodide climatology applying a machine learning approach that combines a compilation of Iodide

observations and other climatologies on parameters such as SST, nitrate, radiation to explain these ob- servations. This compilation of a global ocean water Iodide climatology is of large relevance for large-scale studies on air quality, atmospheric chemistry and climate interactions given the role of ocean Iodide in emissions to the atmosphere affecting atmospheric composition and also involving some potentially relevant feed- back mech- anisms. Overall the paper is well written and presents a sound approach to provide a new dataset to be further applied in Earth system studies and fits within the scope of ESSD. Consequently, I recommend publication of this manuscript in ESSD after the following generally minor comments have been addressed. Note that, since I am not experienced with machine learning methods, my comments are mostly limited to the context of the presented work and the description of the main results coming out of this approach.

We are grateful for the positive comments from the reviewer #2 on the scope, content and general use of our manuscript to the community. We have considered all of the points raised by the reviewer and updated the manuscript as described below. Specific comments Abstract: "simple functions of sea-surface temperature (Chance..)"; I would leave out here the references (generally not included in abstract) and rather state that: "have generally fitted sea-surface iodide observations to relatively simple functions us- ing io- dide proxies such as nitrate and sea-surface temperature"

Updated.

Page 2: line 4, "..oxygen level."

Updated.

Page 2: line 10: I am generally not keen on calling for inclusion of references to my work but since this reference is already included in this ms, the study by Ganzeveld et al. (2009) was also mainly aiming to assess the role of Iodide as one of main reactants in oceanic O3 deposition and the resulting impacts on atmospheric composition.

Updated.

Page 2, line 16: "catalytically destroy ozone (Chameides and Davis, 1980)." Here it would be interesting to add here that this thus mechanisms thus implies the presence of a negative feedback mechanism involving this O3 and Iodide chemistry as also being assessed in a modelling study by Prados Roman et al. (2016)

Added.

Page 4, line 7/8: "they need to be available at an appropriate resolution as a gridded product"; here it would be useful to indicate an estimate of this required resolution given the (known) scale of the heterogeneity in the distribution of the parameters that potentially explain Iodide. For example, given the anticipated (large) contrasts between coastal and open ocean waters, what minimum resolution is needed?

The appropriateness of resolution in this sentence was meant to convey that the input datasets should be as close as possible to the target prediction resolution. The coarseness of prediction resolution is primarily driven by the available resolution of gridded products. This sentence has been updated for clarity.

Page 4; line 14: " This horizontal resolution was used as this is the highest resolution of the current generation of global atmospheric chemistry simulations (Hu et al., 2018)." Here it might be interesting to mention that this resolution of 12.5km also seems to be sufficient for application meso-scale meteorological – Air quality model studies used for regional scale studies. We deploy for example now the meso-scale modelling system WRF-CHEM at a resolution of âĹij20km, including a mechanistic representation of oceanic ozone deposition including Iodide reactivity.

Added.

Section 3.1: Not being very familiar with the application of machine learning methods, I really appreciate the explanation that this given on the specifics of the approach. There is still some terminology that would require further in-depth checking out the details of

the followed approach but think that is a nice way to also explain it all to the readers mostly interested at the end in the final outcome of application of this methodology, the global Iodide dataset.

We thank the reviewer for they positive comment about how we have explained the specifics of the approach taken.

Page 9, line 35: ". . .variability is both well constrained by observations. Some of the highest. . ."

Updated.

Page 10, lines 12-14: " The new predicted values lay between Chance et al. (2014) and MacDonald et al. (2014) in the tropics, however, within the polar regions, the new prediction is significantly higher than both of the previous parameterisations." Not so much a comment but so this result is further stressing the need for additional measurements in the Arctic that we might now get with the upcoming MOSAiC field campaign.

We agree with the reviewers' comment. The importance of additional measurements in the Arctic is now highlighted in the conclusions.

Page 11, line 26/27: "A higher iodide sea-surface concentration would also result in a greater calculated ozone deposition (Luhar et al., 2017; Sarwar et al., 2016).". Here a reference to the Ganzeveld et al. 2009 paper would be really appropriate with this paper showing the first step to consider the impact of global Iodide distribution on global ozone deposition (and atmospheric ozone).

Added reference.

Page 11, line 31: "Considering that the average predicted concentration globally here is 106 nM (Sect. 4.2), these errors are notable"

Updated.

Figure 6: here the observations are indicated by dots that are so small that you cannot

see to what extent the inferred values compare to those observations. You could try to enhance the size of those dots.

The size of circles showing observations were increased, along with clarity of all spatial plots in the manuscript.

Please also note the supplement to this comment:
https://www.earth-syst-sci-data-discuss.net/essd-2019-40/essd-2019-40-AC1-supplement.pdf